# Retromer subunit, VPS29, regulates synaptic transmission and is required for endolysosomal function in the aging brain

Hui Ye[1,2], Shamsideen A Ojelade[2], David Li-Kroeger[1], Zhongyuan Zuo[1], Liping Wang[3], Yarong Li[2], Jessica YJ Gu[4], Ulrich Tepass[4], Avital Adah Rodal[5], Hugo J Bellen[1,3,6,7,8], Joshua M Shulman[1,2,3,6,8]*

[1]Department of Molecular and Human Genetics, Baylor College of Medicine, Houston, United States; [2]Department of Neurology, Baylor College of Medicine, Houston, United States; [3]Program in Developmental Biology, Baylor College of Medicine, Houston, United States; [4]Department of Cell and Systems Biology, University of Toronto, Ontario, Canada; [5]Department of Biology, Brandeis University, Waltham, United States; [6]Department of Neuroscience, Baylor College of Medicine, Houston, United States; [7]Howard Hughes Medical Institute, Houston, United States; [8]Jan and Dan Duncan Neurological Research Institute, Texas Children's Hospital, Houston, United States

**Abstract** Retromer, including Vps35, Vps26, and Vps29, is a protein complex responsible for recycling proteins within the endolysosomal pathway. Although implicated in both Parkinson's and Alzheimer's disease, our understanding of retromer function in the adult brain remains limited, in part because *Vps35* and *Vps26* are essential for development. In *Drosophila*, we find that *Vps29* is dispensable for embryogenesis but required for retromer function in aging adults, including for synaptic transmission, survival, and locomotion. Unexpectedly, in *Vps29* mutants, Vps35 and Vps26 proteins are normally expressed and associated, but retromer is mislocalized from neuropil to soma with the Rab7 GTPase. Further, *Vps29* phenotypes are suppressed by reducing Rab7 or overexpressing the GTPase activating protein, TBC1D5. With aging, retromer insufficiency triggers progressive endolysosomal dysfunction, with ultrastructural evidence of impaired substrate clearance and lysosomal stress. Our results reveal the role of Vps29 in retromer localization and function, highlighting requirements for brain homeostasis in aging.

*For correspondence:
joshua.shulman@bcm.edu

## Introduction

The endolysosomal membrane system comprises a dynamic network of interconnected compartments that mediates sorting or degradation of endocytosed proteins (*Klumperman and Raposo, 2014*; *Repnik et al., 2013*). Many factors that regulate endolysosomal trafficking are implicated in human disease, including neurodegenerative disorders (*Small and Petsko, 2015*; *Soukup et al., 2018*). Among these, retromer is a complex that recycles selected protein cargoes from the endosome to the *trans*-Golgi network or the plasma membrane (*Burd and Cullen, 2014*; *Lucas and Hierro, 2017*). Mutations in *VPS35*—encoding a retromer core protein—are a rare cause of familial Parkinson's disease (*Vilariño-Güell et al., 2011*; *Zimprich et al., 2011*), and retromer has also been linked to endocytic trafficking and processing of the Amyloid Precursor Protein and Tau in Alzheimer's disease (*Muhammad et al., 2008*; *Vagnozzi et al., 2019*; *Wen et al., 2011*). Despite its emerging importance in neurodegenerative disease, the requirements of retromer in neurons and particularly within the aging nervous system remain incompletely understood.

Originally discovered in the yeast *Saccharomyces cerevisiae* (*Seaman et al., 1998*), the heterotrimeric core proteins of the retromer, VPS35, VPS26, and VPS29, are highly conserved among eukaryotes (*Koumandou et al., 2011*). Retromer components are broadly expressed, including in the invertebrate nervous system (*Inoshita et al., 2017*; *Wang et al., 2014*) and throughout the mammalian brain (*Appel et al., 2018*; *Tsika et al., 2014*). In murine neurons, retromer is present in both axons and dendrites, and at the synapse (*Munsie et al., 2015*; *Tsika et al., 2014*). Post-synaptically, retromer appears important for trafficking of AMPA, β2 adrenergic, and perhaps other neurotransmitter receptors to the dendritic membrane (*Choy et al., 2014*; *Munsie et al., 2015*; *Temkin et al., 2017*). In rat mesencephalic cultures, retromer also participates in pre-synaptic dopamine transporter trafficking (*Wu et al., 2017*), and studies of *Vps35* at the *Drosophila* neuromuscular junction suggest a requirement for synaptic vesicle recycling (*Inoshita et al., 2017*). *Vps35*-deficient mice exhibit both defective hippocampal neurotransmission (*Muhammad et al., 2008*) and dopaminergic insufficiency (*Tang et al., 2015a*; *Tang et al., 2015b*), leading to impairments in memory and locomotion, respectively.

Retromer regulates lysosomal degradation pathways, including autophagy, which are important for quality control and brain homeostasis (*Martini-Stoica et al., 2016*; *Menzies et al., 2015*). In the absence of retromer, many protein cargoes are misdirected to the lysosome (*Burd and Cullen, 2014*; *Lucas and Hierro, 2017*; *Wang and Bellen, 2015*), potentially overwhelming degradative capacity, leading to lysosomal expansion and cellular stress. For example, in the *Drosophila* retina, loss-of-function for *Vps35* or *Vps26* results in accumulation of the visual pigment, Rhodopsin-1 (Rh1), within photoreceptors, ultimately causing neuronal dysfunction and loss (*Wang et al., 2014*). Nevertheless, most investigations of the retromer in lysosomal function have relied on cell culture paradigms using non-neuronal cell types (*Cui et al., 2019*; *Jimenez-Orgaz et al., 2018*; *Zavodszky et al., 2014*). Ablation of *VPS35* in the mouse germline is embryonic lethal (*Wen et al., 2011*), and both *Vps35* and *Vps26* similarly have essential developmental requirements in *Drosophila* (*de Vreede et al., 2014*; *Franch-Marro et al., 2008*; *Pocha et al., 2011*; *Starble and Pokrywka, 2018*; *Strutt et al., 2019*; *Wang and Bellen, 2015*).

Notably, among the retromer core proteins, the precise roles of each subunit remain incompletely defined, with especially scant data on VPS29. VPS29 binds the VPS35 C-terminus (*Collins et al., 2005*; *Hierro et al., 2007*; *Kovtun et al., 2018*). Deletion of *Vps29* in yeast or *Caenorhabditis elegans* phenocopies *Vps35* loss-of-function (*Lorenowicz et al., 2014*; *Seaman et al., 1997*). In mammalian epithelial cell culture, reducing VPS29 results in apparent destabilization and degradation of both VPS35 and VPS26 (*Fuse et al., 2015*; *Jimenez-Orgaz et al., 2018*). Reciprocally, pharmacological chaperones targeting the VPS35-VPS29 interface can stabilize the complex and enhance retromer function (*Mecozzi et al., 2014*; *Young et al., 2018*; *Lin et al., 2018*). Here, we have generated and characterized a *Drosophila Vps29* null allele with a focus on in vivo requirements in the nervous system. We identify an unexpected requirement for Vps29 in the regulation of retromer localization, and further highlight a role in synaptic vesicle recycling and lysosomal function in the aging brain.

## Results

### *Vps29* is required for age-dependent retinal function

*Drosophila Vps29* is predicted to encode a 182 amino-acid protein that is 93% similar (83% identical) to human VPS29. Prior studies of *Vps29* in flies have relied on RNA-interference knockdown approaches (*Linhart et al., 2014*). We instead generated a *Vps29* null allele using a CRISPR-Cas9 strategy (*Li-Kroeger et al., 2018*). In the resulting mutant, *Vps29[1]*, the entire coding sequence was replaced by the visible marker gene, $y^{wing2+}$ (*Figure 1A*). Unexpectedly, *Vps29[1]* was homozygous viable, whereas both *Vps35* and *Vps26* mutants are lethal (*Franch-Marro et al., 2008*; *Wang et al., 2014*). Loss of the *Vps29* genomic sequence in null animals was confirmed by PCR (*Figure 1B*) and sequencing of the insertional breakpoints, and we were not able to detect any protein using an anti-Vps29 antibody on western blots from fly head homogenates (*Figure 1C*). Although viable, *Vps29[1]* homozygotes are recovered at ratios below Mendelian expectation (*Figure 1—figure supplement 1A*). We also recovered viable *Vps29[1]* animals lacking both maternal and zygotic protein when crossing homozygous females to

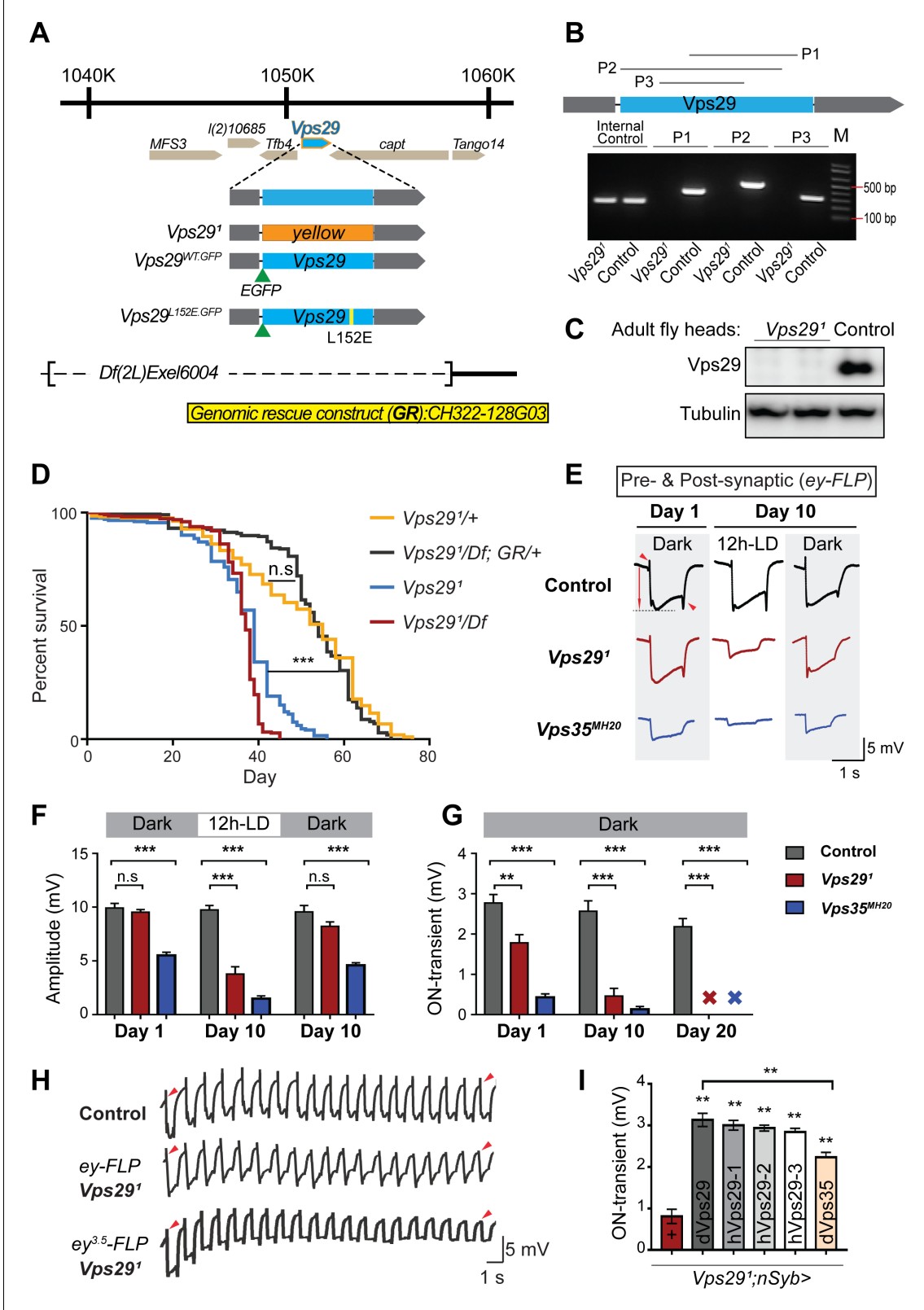

**Figure 1.** *Vps29* is required for age-dependent retinal function. (**A**) The *Vps29* genomic locus is shown, highlighting reagents used in this study. In the null allele, *Vps29¹*, the gene coding sequence (blue, 2L: 1150852–1151420) is replaced by a *ywing²⁺* marker gene. *Vps29^WT.GFP* and *Vps29^L152E.GFP* are identical N-terminal tagged-Vps29 alleles, except for the L152E variant. A chromosomal deficiency *Df(2L)Exel6004* is shown, with the deleted regions indicated by dashed lines. A bacterial artificial chromosome (BAC) (yellow) was used for transgenic genomic rescue (*GR*). (**B**) Genomic polymerase chain

*Figure 1 continued on next page*

*Figure 1 continued*

reaction (PCR) showing loss of Vps29 coding sequence in *Vps29¹* homozygotes versus control (w) flies. P1, P2, and P3 denote expected PCR products from primer pairs targeting *Vps29* genomic sequence. As an additional control, PCR was also performed for *Vps35* genomic sequence. We also performed PCRs using primer pairs that span both sides of the breakpoint junctions abutting the inserted *ywing^(2+)* marker gene cassette (not shown), and these products were Sanger sequenced to confirm the depicted molecular lesion. (C) Western blot from adult heads probed with anti-Vps29 antibody, confirming *Vps29¹* is a protein null allele. (D) *Vps29¹* homozygotes and *Vps29¹/Df* transheterozygotes (*Vps29¹/Df(2L)Exel6004*) show reduced survival that is rescued by the *Vps29* genomic rescue strain (*Vps29¹/Df; GR/+*). Quantitation based on n = 200–235 per group. See also *Figure 1—figure supplement 1A*. (E) Representative ERG traces at 1- and 10 days after generation of *ey-FLP* clones from (i) control (*FRT40A*), (ii) *Vps29¹*, or (iii) *Vps35^MH20*. Flies were raised using an alternating 12 hr light/dark cycle (12h-LD) or in complete darkness. Loss of either *Vps29* or *Vps35* disrupted light-induced depolarization (arrow) in 10-day-old flies. On- and off- transient ERG potentials (top and bottom arrowheads, respectively) were also lost. Raising flies in the dark restored ERG depolarization, but not the transients, indicating persistent defects in synaptic transmission. (F–G) Quantification (n = 6–8) of ERG depolarization and on-transient potentials. 'X' denotes undetectable on-transients in 20-day-old animals. See also *Figure 1—figure supplement 1B*. (H) Compared with controls (*FRT40A* clones), ERG transient potentials are extinguished by rapid stimulation in *Vps29¹* clones. Consistent results were obtained using either *ey-FLP* or *ey^3.5-FLP*, which targets presynaptic neurons only. Flies were raised in complete darkness and examined at 1 day. See also *Figure 1—figure supplement 1C*. (I) Rescue of *Vps29¹* synaptic transmission defects by pan-neuronal expression (*nSyb-GAL4* driver) of either *Drosophila Vps29* or *Vps35* (*dVps29* and *dVps35*) or human *Vps29* (*hVps29-1,−2,* or *−3*, representing three alternate isoforms). Quantification of ERG on-transients in 15-day-old flies (n = 9–12 per group) from the following genotypes: (1) *Vps29¹;nSyb-Gal4/+*; (2) *Vps29¹;nSyb-Gal4/UAS-dVps29*; (3-5) *Vps29¹;nSyb-Gal4/UAS-hVps29*; and (6) *Vps29¹;nSyb-Gal4/UAS-dVps35*. Statistical analysis was based on log-rank test with Bonferroni's correction (D) or one-way ANOVA (F, G, I). All error bars denote SEM. n.s., not significant; **, p<0.01; ***, p<0.001.

The online version of this article includes the following figure supplement(s) for figure 1:

**Figure supplement 1.** Additional studies of the adult retina.

heterozygous males. Notably, *Vps29¹* mutant flies exhibit a modestly reduced survival (~50–60 days versus ~75 days for controls), and this result was confirmed when *Vps29¹* animals were crossed to the deficiency, *Df(2L)Exel6004* (*Figure 1D*). The reduced survival seen in *Vps29* null animals was also rescued by a 23 kb P[acman] bacterial artificial chromosome (*Venken et al., 2009*) containing the *Vps29* genomic locus, establishing specificity.

As introduced above, the retromer plays an essential role in the endocytic recycling of Rh1 in the *Drosophila* eye, thereby enabling proper phototransduction (*Wang et al., 2014*). Consequently, loss of either *Vps35* or *Vps26* leads to aberrant Rh1 trafficking, resulting in lysosomal stress and progressive retinal degeneration. In order to determine if *Vps29* is similarly required, we generated somatic clones of either *Vps29¹* or the *Vps35^MH20* null allele in the fly eye using the *eyeless* (*ey*)-FLP/FRT system (*Newsome et al., 2000*), and monitored degeneration using electroretinograms (ERGs). Compared with control clones (*FRT40A*), removing either retromer component from photoreceptors disrupted light-induced depolarization in 10-day-old animals; however, the *Vps29* phenotype was less severe than that due to loss of *Vps35* (*Figure 1E,F*). Consistent with other retromer components, the *Vps29¹* retinal degeneration phenotype was suppressed when flies were raised in the dark, which eliminates phototransduction and the concomitant endocytosis of Rh1. Age- and light-dependent retinal degeneration in the absence of *Vps29* was further confirmed by histology (*Figure 1—figure supplement 1B*), and this phenotype was also reversed by introduction of the *Vps29* genomic rescue construct. On further inspection of the ERG traces, we noticed that both *Vps29¹* and *Vps35^MH20* appear to disrupt synaptic transmission (*Figure 1E,G*), based on progressive reduction in the on- and off-transient potentials in aging animals (*Babcock et al., 2003*). Interestingly, however, whereas raising animals in the dark rescued phototransduction, the ERG transient potential phenotypes persisted, suggesting a role for retromer at the synapse that is independent of Rh1 recycling.

In our ERG experiments, we also noticed that whereas the transient potentials are preserved in newly eclosed animals with *Vps29* or *Vps35* mutant eye clones, they were poorly sustained throughout the recordings. We therefore repeated our ERG experiments in retinae with either *Vps29¹* or *Vps35^MH20* mutant clones, but instead using a rapid light stimulation paradigm. Indeed, we documented a progressive decline in the on- and off-transient potentials following rapid stimulation, and consistent results were also obtained using the *ey^3.5-FLP* driver (*Mehta et al., 2005*), which selectively targets presynaptic neurons (*Figure 1H* and *Figure 1—figure supplement 1C*). This 'run-down' phenotype is characteristic of mutants that disrupt synaptic vesicle recycling (*Harris and Littleton, 2015*), suggesting that retromer is required for this process. We further demonstrated that the retinal synaptic transmission phenotype in *Vps29¹* homozygotes was fully rescued using the pan-

neuronal expression driver (*nSyb-GAL4*) and a full-length *Drosophila Vps29* cDNA (*nSyb>Vps29*) (*Figure 1I*). Finally, similar rescue was observed when expressing three alternate human *Vps29* isoforms, consistent with functional conservation. Surprisingly, we also found that pan-neuronal overexpression of *Vps35* (*nSyb>Vps35*) partially restored synaptic transmission in *Vps29¹* homozygous animals. In combination with the somewhat weaker phenotype, this result suggests that Vps29 may function primarily to potentiate the activity of other retromer subunits in the retina, such that Vps35 overexpression is compensatory. In sum, whereas Vps29 appears dispensable for retromer function during embryogenesis and development, these studies reveal a requirement in the aging nervous system for phototransduction and synaptic transmission.

## Retromer regulates synaptic vesicle endocytosis and recycling

To further explore the requirement of retromer at synapses, we next turned to the larval neuromuscular junction (NMJ). Based on previous work, Vps35 is present at both the NMJ pre- and post-synapse, and loss of *Vps35* causes synaptic terminal overgrowth and altered neurophysiology suggestive of synaptic vesicle recycling defects (*Inoshita et al., 2017*; *Korolchuk et al., 2007*; *Malik et al., 2015*; *Walsh et al., 2019*). We therefore examined whether *Vps29* mutants show similar phenotypes. Indeed, NMJ preparations from *Vps29¹* homozygous larvae revealed a modest but significant increase in synaptic bouton numbers, consistent with synaptic overgrowth (*Figure 2A*). However, both amplitude and frequency of NMJ miniature excitatory junction potentials (mEJPs) were unaffected following loss of *Vps29* (*Figure 2B*). Similarly, excitatory junction potentials (EJPs) were also preserved (*Figure 2C*). These results suggest that *Vps29* is neither required for spontaneous nor evoked synaptic vesicle release. However, high-frequency stimulation (10 Hz for 10 min) provoked a marked synaptic depression at the *Vps29¹* NMJ (*Figure 2D*). These data provide further support for the role of Vps29 in synaptic vesicle recycling.

The NMJ synaptic run-down phenotype is characteristic of mutants causing defective endocytosis (*Bellen et al., 2010*; *Verstreken et al., 2002*; *Winther et al., 2013*). To further investigate if synaptic vesicle endocytosis requires the retromer, we performed dye uptake studies in animals lacking *Vps29* or *Vps35* (*Figure 2E* and *Figure 2—figure supplement 1*). Larval preparations were stimulated with potassium chloride in the presence of extracellular buffer containing the fluorescent dye FM 1–43, permitting quantification of endocytic flux (*Verstreken et al., 2008*). In control animals, FM 1–43 dye was rapidly internalized at the presynaptic membrane. However, NMJs from either *Vps29¹* or *Vps35^{MH20}* homozygous animals significantly reduced FM dye uptake. These results are consistent with models whereby retromer either directly or indirectly supports vesicle recycling at the presynapse.

## *Vps29* regulates retromer localization and is required in the aging nervous system

The requirement of retromer during *Drosophila* embryogenesis has hindered systematic characterization of its function in the adult nervous system, with the exception of studies in the retina relying on clonal analysis of *Vps35* and *Vps26* (*Wang et al., 2014*). Although *Vps29* is dispensable for embryonic development, our results (above) establish a largely conserved Vps29 requirement for retromer function in the retina and at the NMJ synapse. We therefore studied *Vps29* null adult animals to further explore retromer requirements in the adult brain. Except for the aforementioned retinal requirement, we did not observe obvious morphological defects or apparent evidence of neurodegeneration in the brains of 45-day-old *Vps29¹/Df* adults based on hematoxylin and eosin staining of paraffin sections (*Figure 3—figure supplement 1A*). In order to examine nervous system function, we next tested the startle-induced negative geotactic response (climbing) (*Davis et al., 2016*; *Rousseaux et al., 2018*). Although locomotor behavior was normal in newly eclosed *Vps29* null adults, climbing ability significantly declined with aging (*Figure 3A*). Our results are consistent with published work using RNA-interference to target *Vps29* (*Linhart et al., 2014*). *Vps29¹* homozygous flies manifested a stronger locomotor dysfunction phenotype than *Vps29¹/Df*, suggesting the presence of a potential modifier; however, both genotypes were fully rescued by the genomic BAC transgene. Similar results were obtained for the *Vps29* retinal degeneration phenotype (*Figure 1—figure supplement 1B*). Locomotor defects in *Vps29¹* homozygous animals were also partially rescued by pan-neuronal *Vps29* expression (*nsyb>Vps29*), and human *VPS29* showed comparable

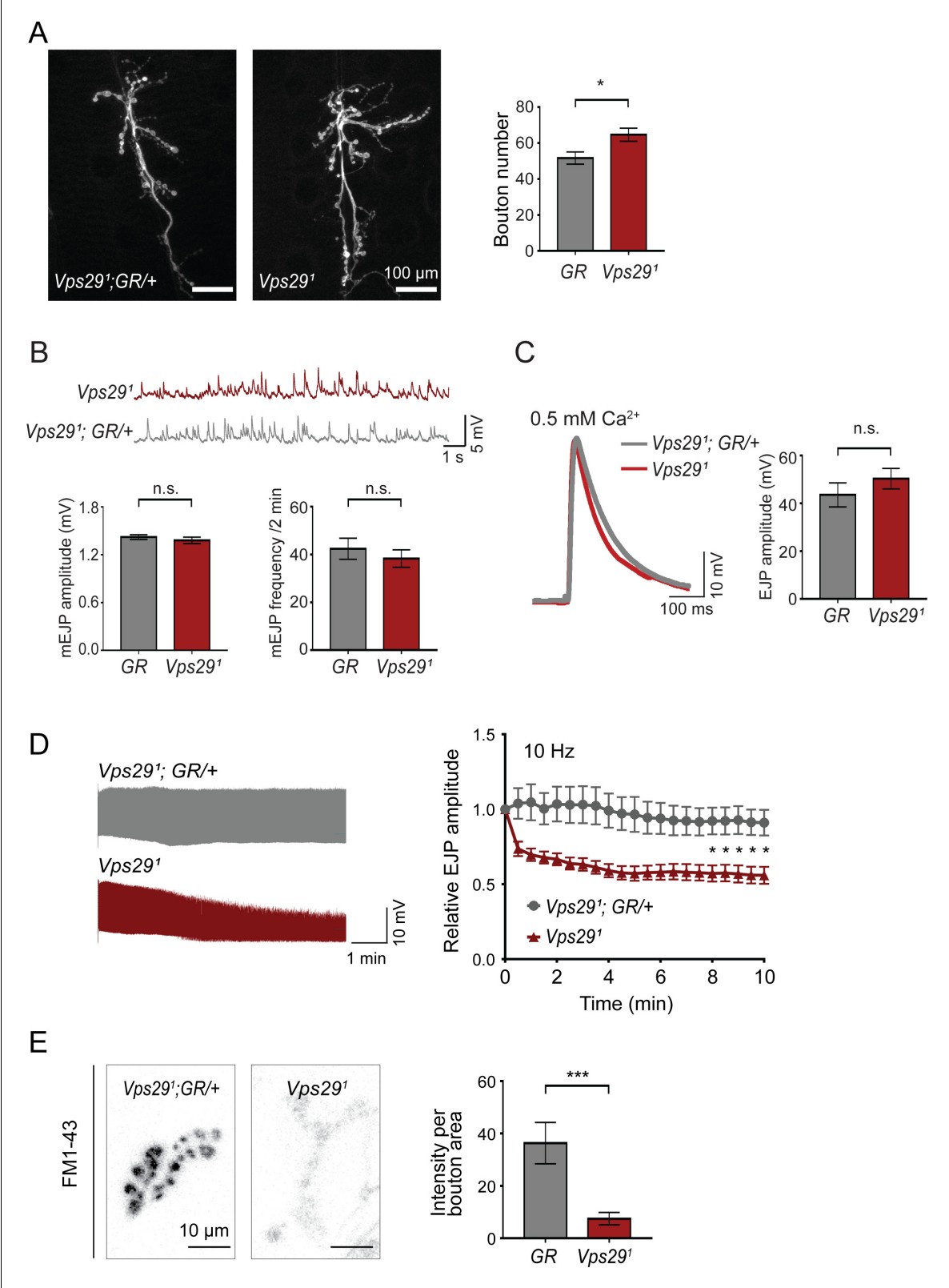

**Figure 2.** Retromer regulates synaptic vesicle endocytosis and recycling. (**A**) *Vps29* loss of function causes an increased number of synaptic terminal boutons at the larval neuromuscular junction (NMJ). NMJ preparations from *Vps29¹* homozygotes or control larvae (*GR* = *Vps29¹;GR/+*) were stained with an antibody against horseradish peroxidase, and Type IIb boutons at abdominal segments A2 and A3 were quantified (n = 5 animals per group). The *Vps29* transgenic BAC (GR) was heterozygous for all comparisons. (**B**) Larval NMJ electrophysiology in *Vps29¹* reveals normal miniature excitatory

*Figure 2 continued on next page*

*Figure 2 continued*

junction potential (mEJP) amplitude and frequency (in 2 min) (n = 14–16). (**C**) Evoked excitatory junction potentials (EJPs) are normal in the absence of *Vps29*. Representative EJP traces from *Vps29¹* and *Vps29¹;GR/+* (control, *GR*) larvae and quantification (n = 13–15). 0.2 Hz stimulation was performed using 0.5 mM Ca²⁺. (**D**) *Vps29¹* NMJs show synaptic depression following rapid stimulation (10 Hz, 0.5 mM extracellular Ca²⁺). Representative traces (10 min) are shown, with quantification (n = 13). Data was normalized to initial EJP amplitude. (**E**) *Vps29¹* NMJs show reduced FM 1-43 dye uptake following KCl stimulation, consistent with impaired synaptic vesicle endocytosis. FM1-43 signal intensity (per bouton area) was quantified (n = 8–11). See also *Figure 2—figure supplement 1*. Statistical analysis (**A–E**) based on Student's t-test. All error bars denote SEM. n.s., not significant; *, p<0.05; ***, p<0.001.

The online version of this article includes the following figure supplement(s) for figure 2:

**Figure supplement 1.** Additional studies of the neuromuscular junction.

rescue activity (*Figure 3B*). Overall, these results suggest *Vps29* is required for the maintenance of nervous system function in aging animals.

In prior studies of HeLa cell cultures, *VPS29* knockdown caused reduced levels of both VPS35 and VPS26, likely due to destabilization of the retromer trimer complex and subunit turnover (*Fuse et al., 2015*; *Jimenez-Orgaz et al., 2018*). Reciprocally, enhancing interactions between VPS29 and VPS35 promotes retromer stability (*Mecozzi et al., 2014*; *Young et al., 2018*). However, Vps35 and Vps26 protein levels were unaffected in *Vps29* null animals (either *Vps29¹* homozygotes or *Vps29¹/Df*), based on western blots from either whole larvae or adult *Drosophila* heads (*Figure 3C* and *Figure 3—figure supplement 1B,C*), including from 30-day-old animals. Furthermore, we confirmed that Vps35 and Vps26 remain tightly associated in the absence of Vps29, based on coimmunoprecipitation assays (*Figure 3D*). Thus, Vps35-Vps26 complex assembly and stability in the nervous system are preserved in *Drosophila* lacking *Vps29*.

Besides expression and stability, retromer function is tightly regulated by its subcellular localization (*Follett et al., 2016*; *Lucas et al., 2016*). We examined retromer expression and localization in the adult fly brain using endogenously tagged fluorescent protein alleles, encoding either Vps35ᴿᶠᴾ or Vps35ᴳᶠᴾ (*Koles et al., 2016*), as well as Vps29ᴳᶠᴾ (*Figure 1A*) fusion proteins. As expected, Vps29 and Vps35 appeared to colocalize, and were broadly expressed in the *Drosophila* brain, including the antennal lobes and the mushroom body (α-/β- lobes and peduncles) (*Figure 4A,B*). We confirmed that most of the observed staining in the adult brain, including within neuropil regions, derived from neuronal Vps35, as this signal was lost following neuronal-specific knockdown (*Figure 4—figure supplement 1A*). However, in *Vps29¹* homozygous animals, Vps35 showed a striking redistribution, shifting from neuropil to enrichment in soma, and forming large perinuclear puncta (*Figure 4C*). Moreover, Vps35 was mislocalized in brains from 1-day-old adults and this result did not significantly change with aging. Consistent results were obtained in the lamina, where retinal photoreceptors synapse on second-order neurons in the visual pathway (*Figure 4—figure supplement 1B*). Specifically, in *Vps29¹* homozygotes, both Vps35 and Vps26 accumulated in the lamina cortex and staining was attenuated in the lamina neuropil. Lastly, we also found that Vps35 was depleted from the *Vps29¹* homozygous larval brain neuropil and at the NMJ synaptic bouton; however, we did not detect any apparent enrichment in motor neuron cell bodies (*Figure 4—figure supplement 2*). Overall, these data suggest that Vps29 is required for normal retromer localization within neurons.

### *Vps29* interacts with *Rab7* and *TBC1D5*

Based on cell culture studies, Vps35 is recruited to endosomal membranes by the Rab7 GTPase (*Seaman et al., 2009*), and *VPS29* knockdown has been previously suggested to increase GTP-bound, activated Rab7 (*Jimenez-Orgaz et al., 2018*). We therefore examined whether dysregulation of Rab7 may similarly participate in neuronal retromer localization in vivo (*Figure 4E–G* and *Figure 4—figure supplement 3*). Indeed, in the adult brain of *Vps29¹* homozygotes, the Rab7 signal appeared strongly increased and was restricted to a somatic, perinuclear pattern with relative depletion from neuropil. The Arl8 GTPase, which functions coordinately with Rab7 in endolysosomal maturation (*Jongsma et al., 2020*; *Marwaha et al., 2017*), was increased in a similar pattern in the *Vps29¹* adult brain. Based on prior studies (*Lund et al., 2018*; *Rosa-Ferreira et al., 2018*), Arl8 localizes to the lysosomal membrane, and we confirmed this using an independent lysosomal marker, Spinster-GFP (*Sweeney and Davis, 2002*). Importantly, mislocalized Vps35 strongly costained with

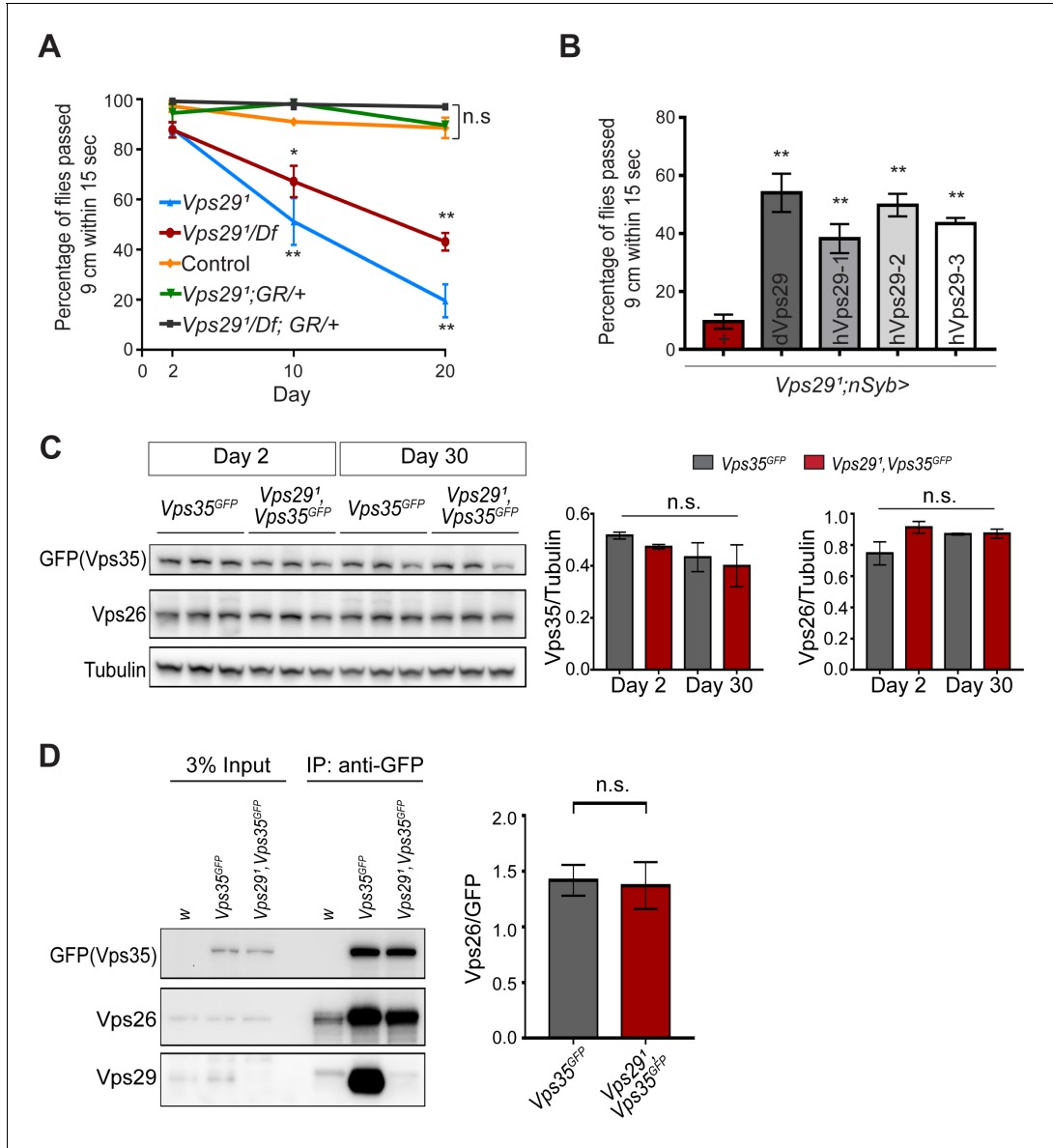

**Figure 3.** Progressive locomotor impairment in *Vps29* mutants but preserved Vps35 and Vps26 protein levels. (**A**) *Vps29[1]/Df* and *Vps29[1]* adults demonstrate age-dependent locomotor impairment, based on startle-induced negative geotaxis, and this phenotype is fully rescued by a single, heterozygous copy of the *Vps29* BAC transgenic (*Vps29[1]/Df; GR/+* and *Vps29[1]; GR/+*). Control: *yw/w*. The *Vps29[1]* homozygote genotype manifests a stronger locomotor phenotype than *Vps29[1]/Df* (p<0.01 in comparisons at 20 days). Quantification based on n = 4 groups, each consisting 14–16 flies. See also *Figure 3—figure supplement 1A*. (**B**) Pan-neuronal expression of either *Drosophila* or human *Vps29* (dVps29 or hVps29, respectively), using the *nsyb-GAL4* driver rescues the *Vps29[1]* locomotor phenotype. Quantification based on n = 4 groups of 20-day-old flies from the following genotypes: (1) *Vps29[1];nSyb-Gal4/+*; (2) *Vps29[1];nSyb-Gal4/UAS-dVps29*; and (3-5) *Vps29[1];nSyb-Gal4/UAS-hVps29*. (**C**) Vps35 and Vps26 protein levels are normal in the absence of *Vps29*. Western blots of adult fly head homogenates from either *Vps29[1], Vps35[GFP]* homozygotes or controls (*Vps35[GFP]*), probed for Vps35[GFP] (anti-GFP), Vps26, or Tubulin (loading control). Quantification based on n = 3 replicate experiments. See also *Figure 3—figure supplement 1B,C*. (**D**) The association of Vps35 and Vps26 proteins is preserved in the absence of *Vps29*. Vps35[GFP] was immunoprecipitated from either *Vps29[1], Vps35[GFP]* homozygotes or control animals (*Vps35[GFP]*) (2 day old), and western blots were probed for Vps35[GFP] (anti-GFP), Vps26, and Vps29. Quantification based on n = 4 replicate experiments. Statistical analysis based on one-way ANOVA (**A–C**) or Student's t- test (**D**). All error bars denote SEM. n.s., not significant; **, p<0.01; ***, p<0.001.

The online version of this article includes the following figure supplement(s) for figure 3:

**Figure supplement 1.** Loss of Vps29 does not affect Vps35 or Vps26 protein expression.

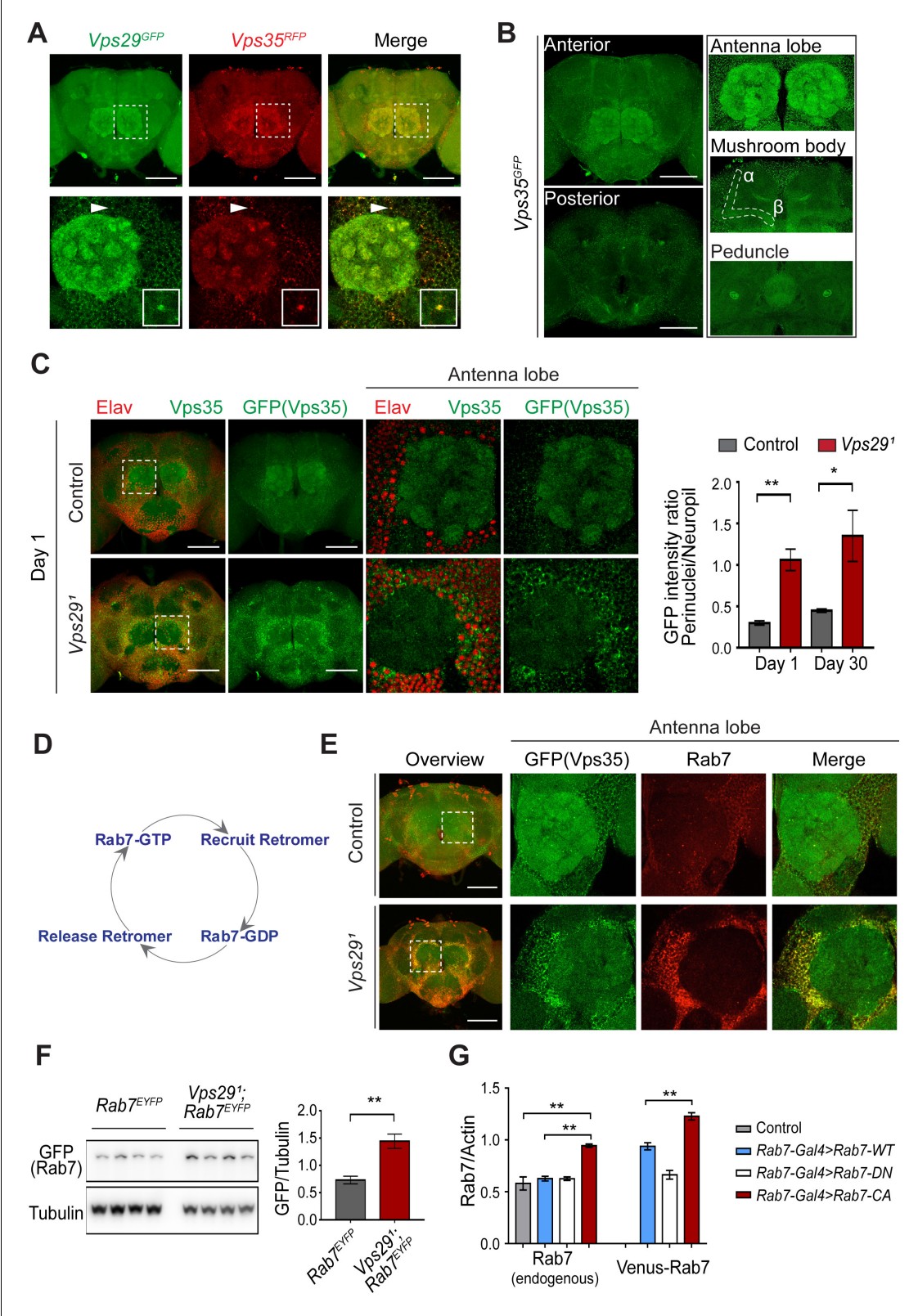

**Figure 4.** *Vps29* regulates Vps35 localization in the adult brain. (**A**) Vps29^GFP (anti-GFP, green) and Vps35^RFP (red) are expressed throughout the adult brain and co-localize. Boxed region of interest including antennal lobe (top row) is shown magnified (bottom row). Representative puncta (arrowhead) co-staining for Vps35 and Vps29 is further magnified in inset. Scale bars, 100 μm. (**B**) Vps35^GFP (anti-GFP, green) is modestly enriched in neuropil regions, including the antennal lobes and the mushroom body (α-/β- lobes and peduncles). Scale bars, 100 μm. See also ***Figure 4—figure supplement***

*Figure 4 continued on next page*

Figure 4 continued

*1A*. (**C**) In the absence of *Vps29*, Vps35$^{GFP}$ (green) protein is redistributed in the adult brain, shifting from neuropil to soma and forming large perinuclear puncta. Neuronal nuclei are labeled with anti-Elav (red). Within the boxed region of interest, including antennal lobe and surrounding nuclei, the GFP intensity ratio (perinuclear to neuropil) was quantified (n = 3) in *Vps29$^1$, Vps35$^{GFP}$* homozygotes or *Vps35$^{GFP}$* controls. Scale bars, 100 µm. See also *Figure 4—figure supplement 1B* and *Figure 4—figure supplement 2*. (**D**) Schematic highlighting the Rab7 cycle including GTP-bound (active) and GDP-bound (inactive) forms and putative interactions regulating retromer recruitment and release. (**E**) In *Vps29* mutants, Vps35$^{GFP}$ (green) colocalizes with Rab7 (anti-Rab7, red), which appears increased and similarly redistributed from neuropil to soma. The region of interest centered on the antenna lobe is indicated in the boxed region of the lower power image (maximal intensity projection); the magnified images are representative anterior single sections taken through the antennal lobe. Scale bars, 100 µm. See also *Figure 4—figure supplement 3*. (**F**) Rab7 protein levels are increased in the absence of *Vps29*. Western blots of adult head homogenates, including *Vps29$^1$, Rab7$^{EYFP}$* homozygotes or *Rab7$^{EYFP}$* controls (2-day-old), were probed for Rab7-YFP (anti-GFP) and Tubulin (loading control). Quantification based on n = 4 replicate experiments. See also *Figure 4—figure supplement 4A*. (**G**) Rab7 protein levels are increased in the constitutive active, 'GTP-locked' *Rab7$^{Q67L}$* mutant. Western blots of adult head homogenates from 5-day-old animals were probed for Rab7 or Actin, including the following genotypes: (1, *Rab7-Gal4 > Rab7* CA) *UAS-Venus-Rab7$^{Q67L}$/+; Rab7-Gal4/+*; (2, *Rab7-Gal4 > Rab7* WT) *UAS-Venus-Rab7$^{WT}$/+; Rab7-Gal4/+*; (3, *Rab7-Gal4 > Rab7* DN) *UAS-Venus-Rab7$^{T22N}$/+; Rab7-Gal4/+*; (4, control) *w*. Venus-tagged or endogenous Rab7 protein levels were separately quantified, based on n = 4 replicate experiments. See also *Figure 4—figure supplement 4B*. Statistical analysis was based on Student's t-test (**C, F**) or one-way ANOVA (**G**). To address the possibility of a modestly skewed distribution, the data in **C** were log$_2$-transformed, and the results of t-tests were unchanged. All error bars denote SEM. *, p<0.05; **, p<0.01. The online version of this article includes the following figure supplement(s) for figure 4:

**Figure supplement 1.** Loss of Vps29 causes mislocalization of retromer in adult brain.
**Figure supplement 2.** Loss of Vps29 causes mislocalization of Vps35 in larvae.
**Figure supplement 3.** Loss of Vps29 causes mislocalization of Rab7.
**Figure supplement 4.** Increased Rab7 expression following loss of *Vps29*.

perinuclear Rab7 and Arl8, suggesting enrichment at the late endosome and/or lysosome. On western blots, we documented an increase in Rab7 protein in either *Vps29$^1$* homozygotes or *Vps29$^1$/Df* (*Figure 4F* and *Figure 4—figure supplement 4A*). To examine if the change in Rab7 expression level reflects an altered ratio of GTP-bound (active) to GDP-bound (inactive) Rab7 protein (*Figure 4D*), we leveraged available fly strains harboring Venus-tagged *Rab7* transgenes (*Cherry et al., 2013*), including wildtype *Rab7*, the 'GTP-locked' mutant *Rab7$^{Q67L}$* and the 'GDP-locked' mutant *Rab7$^{T22N}$*, under the control of identical *Rab7* promoter elements (*Rab7-GAL4*). We found that the constitutively active, GTP-locked Rab7$^{Q67L}$ protein was present at higher levels (*Figure 4G* and *Figure 4—figure supplement 4B*). Moreover, endogenous wildtype Rab7 was also increased, potentially reflecting a compensatory cellular response. These data support a model in which loss of *Vps29* promotes Rab7 activation, redistribution of active Rab7 from axon to soma, and subsequent recruitment of Vps35.

If Rab7 activation mediates the disruption in retromer localization and function in *Vps29* mutants, our model predicts that reduction in Rab7 may rescue the neuronal requirements for Vps29. Indeed, although complete loss-of-function for *Rab7* is lethal, removing one copy (*Rab7$^{Gal4-KO}$/+*) (*Cherry et al., 2013*) partially rescued the age-dependent locomotor impairment and synaptic transmission defects manifest in *Vps29$^1$/Df* animals (*Figure 5A,B*). Reduction of *Rab7* also restored synaptic vesicle endocytosis at the larval NMJ in *Vps29$^1$* homozygotes, based on the FM 1–43 dye uptake assay (*Figure 5C*). In mammalian cell culture studies, VPS29 regulates Rab7 activity via recruitment of the GTPase-activating protein (GAP), TBC1D5 (*Jia et al., 2016*; *Seaman et al., 2009*). *Drosophila* has a single ortholog of TBC1D5 that is well-conserved (28% identity/47% similarity) (*Gramates et al., 2017*). Consistent with our model, pan-neuronal overexpression of *dTBC1D5* (*nsyb>dTBC1D5*) normalized Rab7 protein levels (*Figure 5D*) and partially rescued the synaptic transmission defect in *Vps29* mutants (*Figure 5E*). Next, we introduced a point mutation at the *Vps29* genomic locus causing the single amino-acid substitution, *Vps29$^{L152E}$* (*Figure 1A*); mutation of this conserved residue has been shown to disrupt the interaction with TBC1D5 in mammalian cells (*Jia et al., 2016*; *Jimenez-Orgaz et al., 2018*). *Vps29$^{L152E}$* failed to complement *Vps29$^1$* (*Figure 5F–H* and *Figure 5—figure supplement 1*), causing age-dependent photoreceptor synaptic transmission defects and locomotor impairment. The residual, albeit weak on-transient ERG potential in *Vps29$^1$/Vps29$^{L152E}$* adults suggest that *Vps29$^{L152E}$* is a hypomorphic allele. Moreover, Vps29$^{L152E}$ was expressed at higher levels and Rab7 was also increased, as in *Vps29$^1$*. Overall, these results indicate that Vps29 regulates retromer localization coordinately with Rab7 and TBC1D5.

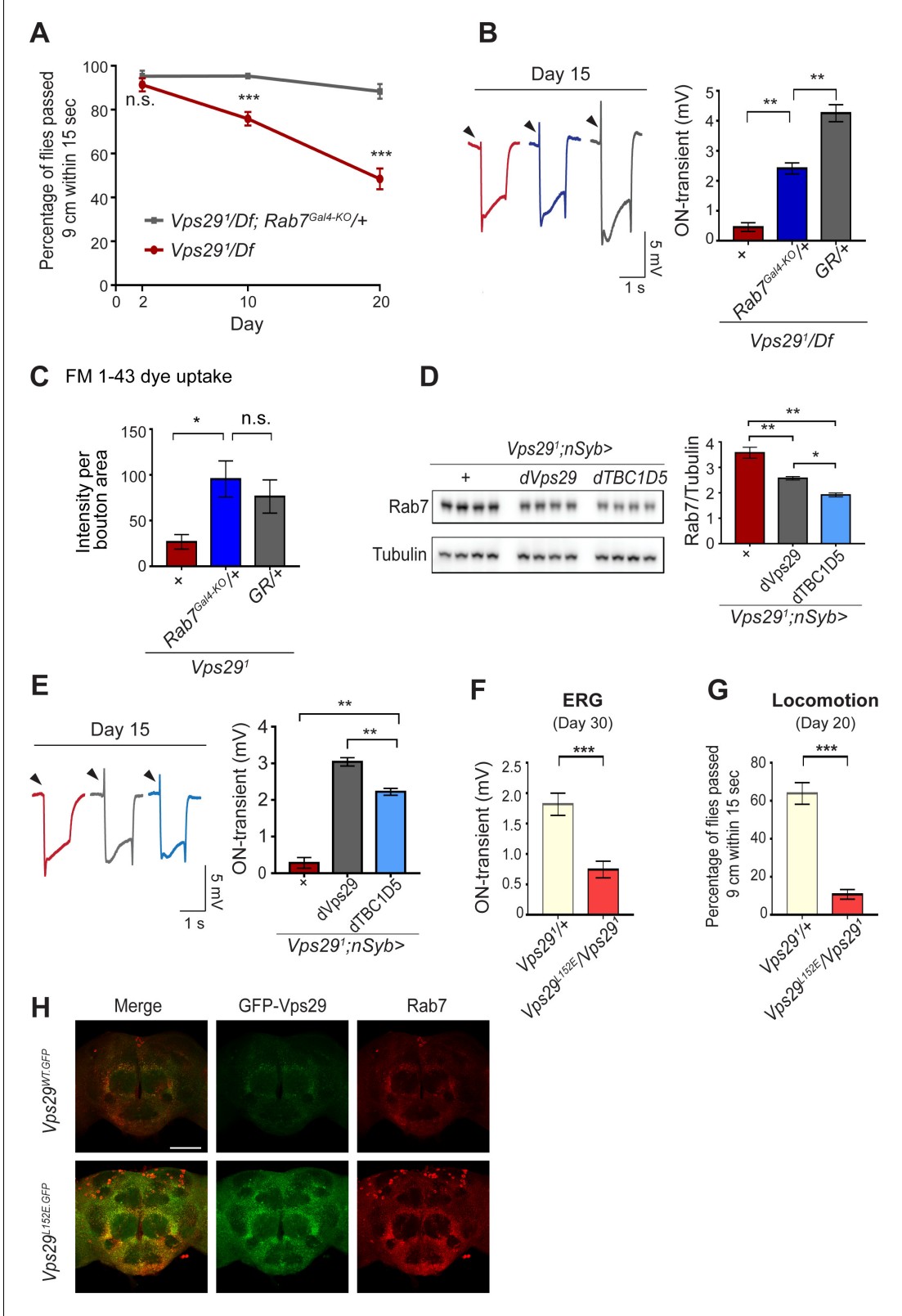

**Figure 5.** Reduction of Rab7 or overexpression of TBC1D5 suppresses *Vps29* mutant phenotypes. (**A–C**) Reduction of *Rab7* rescues *Vps29* mutant phenotypes, including progressive locomotor impairment (**A**), synaptic transmission (**B**), and synaptic vesicle endocytosis at the larval NMJ (**C**). (**A**) Quantification of locomotor behavior based on n = 4 groups, each consisting of 12–15 flies. (**B**) Quantification of electroretinogram (ERG) on-transients (n = 12–13) from adults raised using a 12 hr light/dark cycle. (**C**) FM 1-43 dye uptake signal intensity per NMJ bouton area was quantified (n = 7–9). (**D**)

*Figure 5 continued on next page*

*Figure 5 continued*

Pan-neuronal overexpression of *dTBC1D5* (*nsyb>dTBC1D5*) restores Rab7 protein level in flies lacking *Vps29*. Western blots of adult fly head homogenates (30-day-old) were probed for Rab7 and Tubulin (loading control), and quantified based on n = 4 replicate experiments, including the following genotypes: (1, red) *Vps29¹;nSyb-GAL4/+*; (2, grey) *Vps29¹;nSyb-GAL4/UAS-dVps29*; (3, blue) *Vps29¹;nSyb-GAL4/UAS-dTBC1D5*. (E) Pan-neuronal overexpression of *dTBC1D5* (*nsyb>dTBC1D5*) rescues synaptic transmission in *Vps29¹* mutants. Quantification of ERG on-transients (n = 8) from adults raised using a 12 hr light/dark cycle. (F, G) The $Vps29^{L152E}$ mutation, predicted to disrupt the Vps29-TBC1D5 interaction, fails to complement *Vps29¹*, causing impaired synaptic transmission (F) and locomotor impairment (G). Quantification of ERG on-transients in 30-day-old flies raised in complete darkness (F) (n = 12–14) or locomotor behavior in 20-day-old flies (G) (n = 5–6 groups). See also *Figure 5—figure supplement 1A*. (H) The $Vps29^{L152E}$ mutation causes increased expression of Vps29 (anti-GFP, green) and Rab7 (anti-Rab7, red). Whole-mount brain immunofluorescence shown for $Vps29^{L152E.GFP}$ or $Vps29^{WT.GFP}$ homozygotes (2-day-old adults). Scale bars, 100 µm. See also *Figure 5—figure supplement 1B*. Statistical analysis based on Student's t-test (A, F, G) or one-way ANOVA (B–E). All error bars denote SEM. n.s., not significant; *, p<0.05; **, p<0.01; ***, p<0.001.

The online version of this article includes the following figure supplement(s) for figure 5:

**Figure supplement 1.** Additional characterization of the $Vps29^{L152E}$.

## Loss of *Vps29* causes lysosomal dysfunction during aging

As introduced above, retromer mediates the recycling of protein cargoes from the endolysosomal system to the *trans*-Golgi network or plasma membrane. Moreover, retromer loss may impair lysosomal biogenesis and function, for example by disrupting delivery of key hydrolases (*Maruzs et al., 2015*). In the *Drosophila* retina, besides aberrant trafficking of Rh1 (*Wang et al., 2014*), loss of *Vps35* causes accumulation of ceramide lipid species (*Lin et al., 2018*), which contributes to lysosomal expansion, cellular stress, and ultimately photoreceptor loss. We confirmed that glucosylceramide similarly accumulates in *Vps29¹* homozygous photoreceptors (*Figure 6A*), consistent with the retinal degeneration phenotype. In order to better understand the proximal consequences of retromer mislocalization that lead to neuronal dysfunction, we performed a series of experiments to assay lysosomal function in the adult brains of *Vps29* mutants. We first examined the lysosomal proteases, cathepsin D (CTSD) and cathepsin L (CTSL). These enzymes are each synthesized as propeptides in the endoplasmic reticulum (ER) and become active following cleavage in the acidic, lysosomal milieu. Lysosomal dysfunction therefore commonly leads to reduced levels of mature cathepsins and increased levels of the immature proforms (*Hasanagic et al., 2015*; *Li et al., 2017*). We detected normal levels of mature CTSL and increased mature CTSD in adult head homogenates from 1-day-old *Vps29¹/Df* animals (*Figure 6B* and *Figure 6—figure supplement 1A*). However, in 30-day-old animals, we found that the uncleaved proforms of both cathepsins were sharply increased, consistent with progressive lysosomal dysfunction with aging.

We next examined well-established markers of lysosomal autophagy, which plays a critical role in nervous system homeostasis and neurodegeneration (*Martini-Stoica et al., 2016*; *Menzies et al., 2015*). Substrate degradation is triggered when Atg8-decorated autophagosomes fuse with lysosomes, leading to turnover of Ref(2)P/p62 and concomitant clearance of polyubiquitinated proteins. We did not observe any accumulation of these markers in heads from either 1- or 30-day-old *Vps29¹/Df* flies (*Figure 6—figure supplement 1B,C*), consistent with preserved autophagic flux. Interestingly, however, at more extreme ages (45 days), *Vps29* mutants showed evidence of accelerated autophagic failure, with more significant elevation of both p62 and Atg8, and accumulation of polyubiquitinated proteins (*Figure 6C*). Notably, reduction of *Rab7* (*Rab7^{Gal4-KO}/+*) improved autophagic clearance in *Vps29* mutant brains.

Lastly, in order to directly visualize lysosomal ultrastructure, we performed transmission electron microscopy (TEM) on retinae and brains from 30-day-old *Vps29¹/Df* animals (*Figure 7* and *Figure 7—figure supplement 1*). For the retinal studies, we selected conditions for which photoreceptor depolarization is preserved, preceding overt retinal neurodegenerative changes. Indeed, retinal TEM revealed normal numbers and morphology of photoreceptors; however, we documented significantly increased numbers of lysosomes, multivesicular bodies, and autophagic vacuoles. Further, lysosomes were frequently aberrantly enlarged and filled with granular, electron-dense material. These findings are highly suggestive of endolysosomal dysfunction in *Vps29¹/Df* animals, in agreement with results from other assays. Interestingly, although synaptic transmission is disrupted in these animals (i.e. reduced ERG on/off transients), *Vps29* mutant synaptic terminals in the lamina appeared unremarkable. We observed normal synapse numbers and size, nor did we detect any

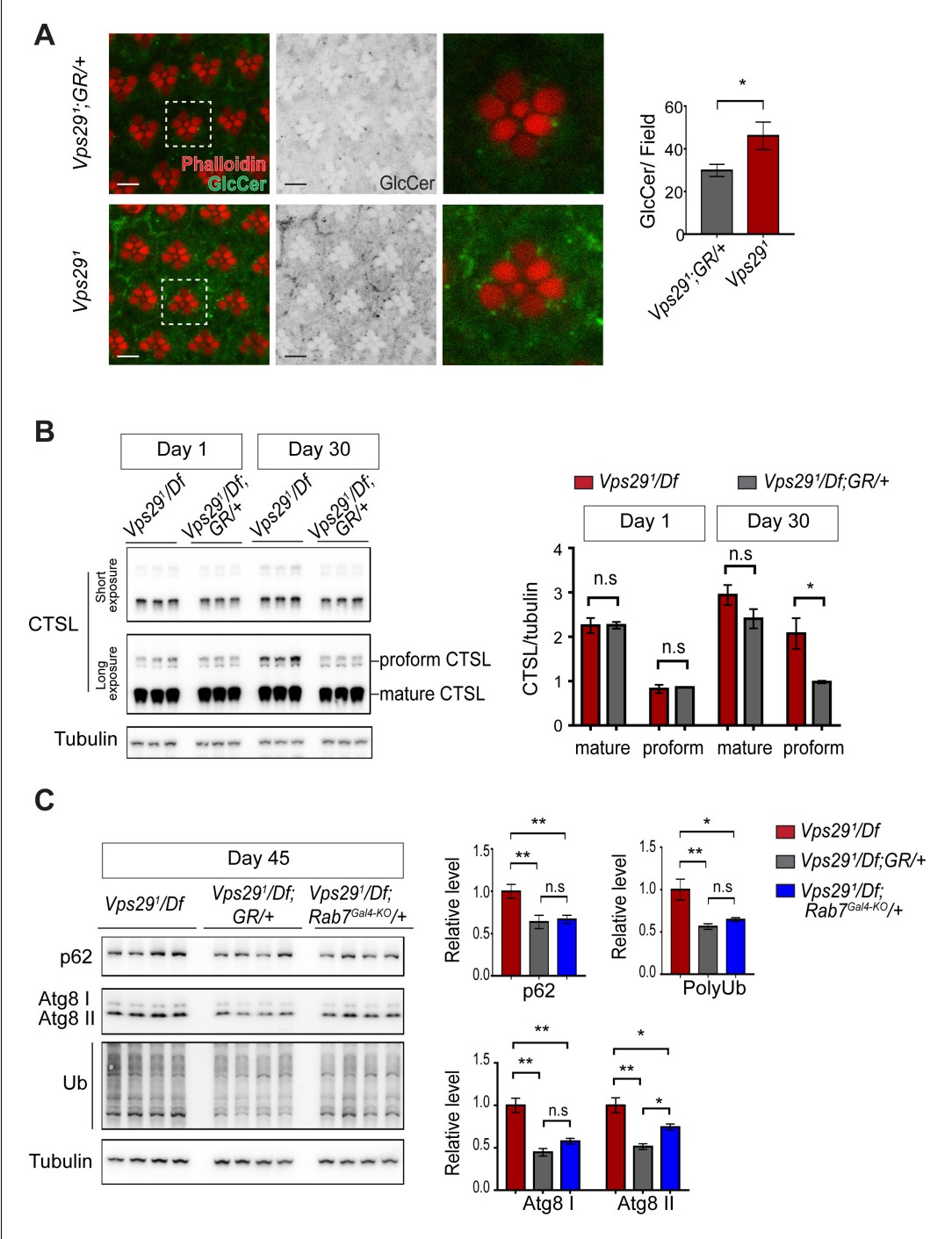

**Figure 6.** Loss of *Vps29* causes age-dependent, progressive lysosomal dysfunction in the brain. (**A**) In the absence of *Vps29*, glucosylceramide (anti-GlcCer, green) accumulates in the retina. Tissue is counterstained for actin (phalloidin, red) to highlight the photoreceptor rhabdomeres. Glucosylceramide signal was quantified from a $36.94 \times 36.94$ µm$^2$ region encompassing ~10 ommatidia in 30-day-old animals (n = 10) raised in dark conditions, including *GMR-w-RNAi/+; Vps29¹; GR/+* (controls) and *GMR-w-RNAi/+; Vps29¹*. Eye pigment was removed using RNAi against the *w* gene to reduce auto-fluorescence. A representative ommatidium (boxed) is shown at higher-power (right). Scale bars, 5 µm. (**B**) The uncleaved, CTSL proform is increased in aged animals lacking *Vps29*, consistent with diminished lysosomal proteolytic capacity. Western blots from adult fly head homogenates were probed for CTSL and Tubulin (loading control), and quantified based on n = 3 replicate experiments. See also **Figure 6—figure supplement 1A**. (**C**) Autophagic flux is reduced in aged animals lacking *Vps29*, and this phenotype is suppressed by reduction of *Rab7*. Western blots of adult head homogenates were probed for autophagic markers, including p62, Atg8, polyubiquitin (FK1), or Tubulin (loading control), and quantified based on

*Figure 6 continued on next page*

*Figure 6 continued*

n = 8 replicate experiments. See also *Figure 6—figure supplement 1B,C*. Statistical analysis based on Student's t-test (**A, B**) and one-way ANOVA (**C**). All error bars denote SEM. n.s., not significant; *, p<0.05; **, p<0.01.

The online version of this article includes the following figure supplement(s) for figure 6:

**Figure supplement 1.** Additional studies of lysosomal proteolysis and autophagy.

other apparent evidence of synaptic degeneration. Lastly, we examined TEM of adult brain sections in 30-day-old animals, focusing on cortical regions with densely packed neuronal cell bodies in the dorsal-posterior brain. Consistent with findings in the retina, we discovered significantly increased numbers of aberrant lysosomal structures (multilamellar bodies) in *Vps29* mutants. Collectively, our data indicate that retromer dysfunction following loss of *Vps29* is associated with progressive lysosomal structural and functional degeneration in the aging nervous system.

## Discussion

Although strongly implicated in pathogenesis of human neurodegenerative disease, the requirements for retromer in the aging brain remain poorly defined. Our discovery that Vps29 is dispensable for *Drosophila* embryogenesis provides an unexpected opportunity to examine the neuronal consequences of retromer dysfunction in vivo. In the absence of *Vps29*, we document age-dependent impairment in lysosomal proteolysis and autophagy, leading to apparent accumulation of undigested substrates and expansion of the endolysosomal compartment. These cellular changes are accompanied by progressive nervous system dysfunction, with synaptic transmission especially vulnerable. Unexpectedly, Vps29 loss did not affect the expression or stability of other retromer components but influenced their localization via a Rab7- and TBC1D5-dependent regulatory pathway.

### VPS29 is required for retromer recruitment via Rab7 and TBC1D5

The retromer core initiates cargo recognition and mediates the engagement of other factors that enable proper endocytic trafficking (*Cullen and Steinberg, 2018*). However, the differential requirements of each protein within the heterotrimer, including VPS35, VPS26, and VPS29, remain incompletely defined. Structural studies suggest the retromer core has an elongated, flexible structure, with VPS35 serving as a central scaffold, binding VPS26 at its N-terminus, and VPS29 at its C-terminus (*Hierro et al., 2007*; *Kovtun et al., 2018*; *Lucas et al., 2016*). Consistent with this, loss of VPS35 destabilizes the retromer complex, resulting in reduced VPS26 and VPS29 protein, and this result has been confirmed both in vitro and in vivo (*Cui et al., 2019*; *Liu et al., 2014*; *Wang et al., 2014*). Similarly, knockdown or knockout of *VPS29* in mammalian epithelial cell culture compromises retromer stability, causing loss of the other components (*Fuse et al., 2015*; *Jimenez-Orgaz et al., 2018*). Surprisingly, in the absence of Vps29 in *Drosophila*, we found that Vps35 and Vps26 are present at normal levels and remain tightly associated. Instead, Vps35 appears to be mislocalized, shifting from neuropil to soma in the adult brain, and likely disrupting its normal function. Loss of Vps29 also causes apparent hyperactivation (and mislocalization) of Rab7, and a mutation previously shown to disrupt the VPS29-TBC1D5 interaction shows similar phenotypes. Importantly, reduction of Rab7 or overexpression of TBC1D5 potently suppresses *Vps29* loss-of-function. In the adult brain, perinuclear accumulation of Vps35/Rab7 strongly suggests aberrant localization to the late endosome and lysosome, based on the overlap with Arl8 and Spinster. Our in vivo findings recapitulate a two-step molecular mechanism for retromer recruitment and release in neurons (*Figure 8*). First, the retromer core is recruited to the endosomal membrane by Rab7-GTP, allowing it to participate in cargo recycling. Subsequently, Vps29 engages TBC1D5, which activates GTP hydrolysis of Rab7, thereby releasing retromer from the endosome. We propose that in the absence of Vps29, retromer is

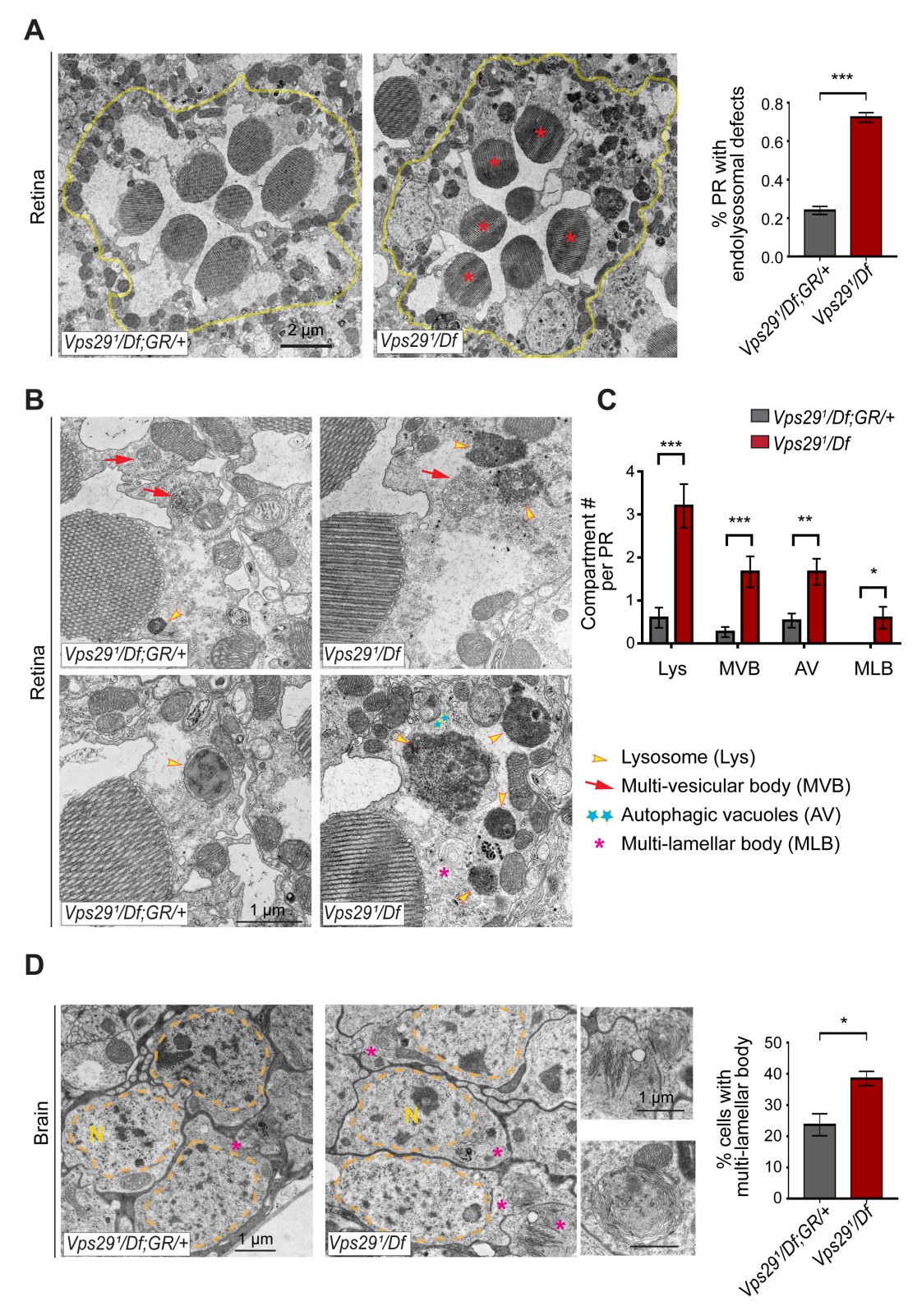

**Figure 7.** Loss of *Vps29* disrupts lysosomal ultrastructure in the brain. (**A**) Transmission electron microscopy (TEM) reveals overall preserved photoreceptor morphology, but aberrant endolysosomal structures in retinae from 30-day-old animals lacking *Vps29*. Within each ommatidium (yellow outline), the percentage of photoreceptors (PR) with endolysosomal defects was quantified. Asterisks denote photoreceptors with aberrant endolysosomal structures. Quantification based on n = 30 ommatidia from three animals per genotype. See also *Figure 7—figure supplement 1A,B*.

*Figure 7 continued on next page*

*Figure 7 continued*

(B) At higher magnification, TEM reveals an increase in lysosomes, multivesicular bodies, autophagic vacuoles, and multilamellar bodies in *Vps29* mutant retinae. Lysosomes were frequently observed to be aberrantly enlarged and filled with granular, electron-dense material. (C) Distinct endolysosomal structures and/or compartments were quantified in n = 15 photoreceptors, including five photoreceptors from three animals for each genotype. (D) Increased numbers of multi-lamellar bodies (asterisks) are observed in cortical neurons from brains 30-day-old animals lacking *Vps29*. Neuronal nuclei ('N') are outlined. Quantification based on cell counts from n = 4 animals (50 cells per brain). See also *Figure 7—figure supplement 1D*. Statistical analysis (A, C, D) based on Student's t-test. All error bars denote SEM. *, p<0.05; **, p<0.01; ***, p<0.001.

The online version of this article includes the following figure supplement(s) for figure 7:

**Figure supplement 1.** Additional ultrastructural analysis of *Vps29* mutants.

trapped at the endosome membrane in the neuronal cell body, leading to sequestration of functional complexes, and progressive impairment in endocytic trafficking.

The reciprocal interplay and interdependence of retromer and Rab7 that we document in the *Drosophila* nervous system is consistent with prior studies in cell culture (*Jimenez-Orgaz et al., 2018*; *Liu et al., 2012*; *Seaman et al., 2009*). By contrast, the viability of *Vps29* mutants as well as our finding of a more selective, regulatory role for Vps29 in complex localization is unexpected. Unlike the other retromer core subunits, *Vps29* is dispensable for embryogenesis and loss-of-function causes milder retinal phenotypes than *Vps35* or *Vps26*. Moreover, Vps35 overexpression partially rescued selected *Vps29* phenotypes, consistent with a context-dependent, regulatory role. Since Vps35 can weakly bind TBC1D5 (*Jia et al., 2016*), it may be capable of inactivating Rab7, compensating in part for the absence of Vps29. Our finding that retromer

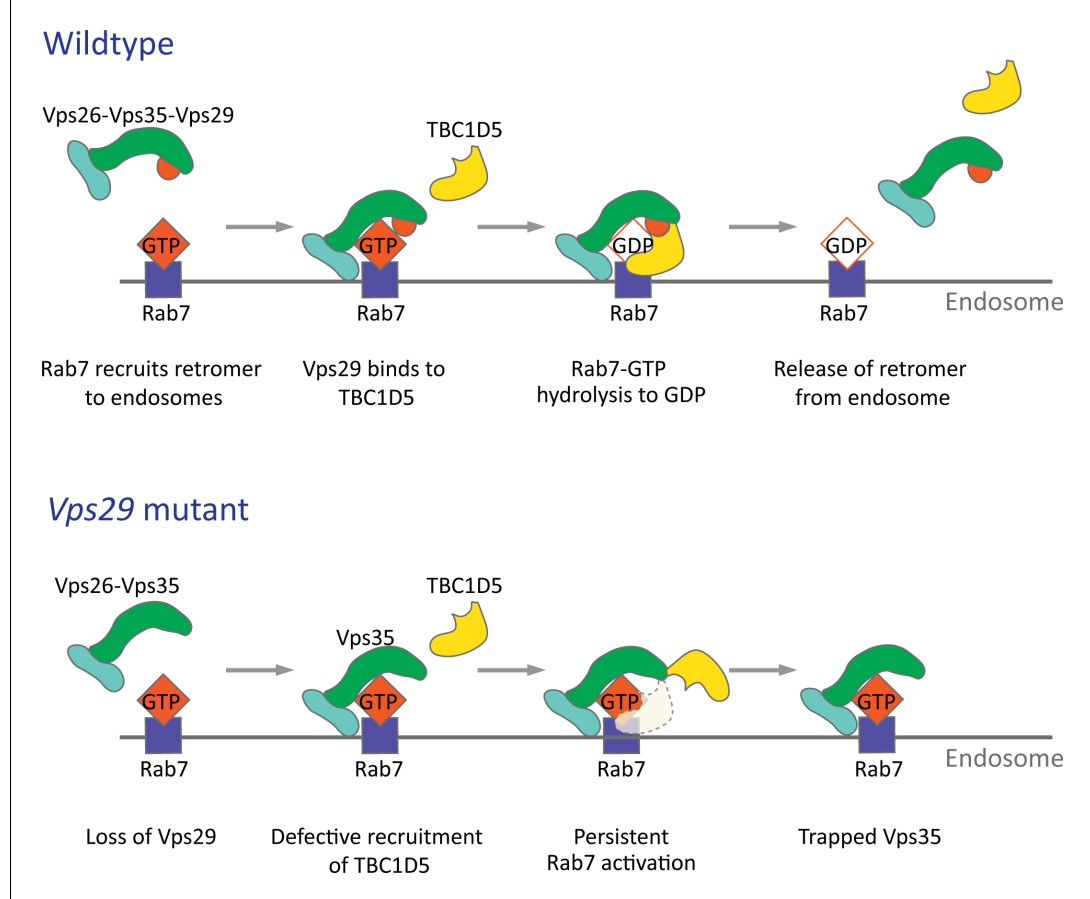

**Figure 8.** Model for Vps29-dependent retromer recruitment and release. In neurons, the retromer core Vp26-Vps35-Vps29 is recruited by Rab7 to the endosomal membrane. Vps29 engages TBC1D5, which promotes inactivation of Rab7 and release of Vps35. In the absence of *Vps29*, the residual retromer complex is trapped at the endosomal membrane.

retains some residual activity in flies lacking *Vps29* is reminiscent of results from *C. elegans* (*Lorenowicz et al., 2014*). Moreover, in *S. cerevisiase*, interaction of Vps26 and Vps35 are similarly preserved following loss of *Vps29* (*Reddy and Seaman, 2001*). Importantly, these data suggest that the specific requirement(s) for Vps29—and by extension, retromer activity—may be context dependent, varying with cell-type (e.g. neuron) and/or specific cargo (e.g. Rh1, ceramides, or others). While *Vps29* mutants altered Vps35 subcellular localization in both the adult and larval brains causing relative loss from neuropil regions, potential differences were also noted: the apparent redistribution to soma was only seen in adults. It is also possible that tissue- or species-specific differences might relate to Vps29 participation in other endosomal sorting machinery besides retromer, such as the recently described 'retriever' (*McNally et al., 2017*); however, this complex has not yet been studied in *Drosophila*. Although we have found that human *VPS29* is capable of functional substitution in flies, it will be important to determine whether our findings also apply in the mammalian nervous system context.

## The retromer and endolysosomal trafficking in neurons

Our characterization of flies lacking *Vps29* points to important roles for retromer in both lysosomal and synaptic homeostasis in neurons, interdependent processes that similarly require proper endocytic trafficking. Following *Vps29* loss-of-function, we detected evidence of progressive deterioration in lysosomal function in the adult brain, based on markers of cathepsin maturation and autophagic flux. In both cases, lysosomal function was preserved in newly eclosed animals, but deficits emerged gradually with aging. This result suggests that neurons can initially compensate for retromer insufficiency, whereas adaptive mechanisms may breakdown during aging. Vps35 was strongly mislocalized in 1-day-old animals, suggesting that the manifestation of age-dependent vulnerability likely involves more downstream consequences of retromer dysfunction, such as the accumulation of undigested substrates and resulting lysosomal expansion. Indeed, we found ultrastructural evidence of lysosomal stress in the aged retina and brain, including significantly enlarged lysosomes that were filled with electron-dense and/or multilamellar material. One caveat for interpretation is that TEM data was not included for newly eclosed animals. Nevertheless, our results are consistent with evidence from other experimental systems, that protein and lipid cargoes accumulate gradually in the late endosome and lysosomes in the absence of retromer activity (*Lin et al., 2018*; *Tang et al., 2015a*; *Wang et al., 2014*; *Wen et al., 2011*). Neuronal compensation may entail increasing capacity for substrate turnover, since we found that mature forms of CTSD and markers of autophagic flux were increased in young animals (*Figure 6—figure supplement 1*). It is also possible that the production of critical membrane proteins dependent on retromer for recycling can be boosted, at least temporarily. The initial resilience of the brain to *Vps29* deficiency that we uncover in *Drosophila* is consistent with relative preservation of autophagy following VPS29 knockout in HeLa cells (*Jimenez-Orgaz et al., 2018*). However, there is ample evidence from animal model studies that aging is accompanied by reduced degradative capacity in neurons, which may ultimately contribute to decompensation and increased proteotoxic stress (*Mattson and Magnus, 2006*).

We additionally found that *Vps29* is required for synaptic transmission in both the *Drosophila* retina and larval NMJ, producing a phenotype characteristic of other genes that regulate endocytosis and synaptic vesicle recycling (*Bellen et al., 2010*; *Harris and Littleton, 2015*). In *Drosophila* S2 cells, Vps35 has been shown to participate in endocytosis (*Korolchuk et al., 2007*), and our findings confirm and extend prior work (*Inoshita et al., 2017*) implicating Vps35 in synaptic structure and function at the NMJ. Whereas both genes appear similarly required for synaptic vesicle endocytosis, loss of *Vps35* but not *Vps29*, also affected spontaneous NMJ activity, causing increased miniature excitatory junction potentials (*Inoshita et al., 2017*). This is consistent with our other data suggesting *Vps29* causes overall milder defects. *Vps29* synaptic phenotypes were suppressed by either reduction of Rab7 or overexpression of TBC1D5, similar to our studies of lysosomal phenotypes. Interestingly, ERGs also revealed age-dependent, progressive decline in synaptic transmission. Despite evidence of significant degradation of synaptic function with aging, TEM studies of the *Drosophila* lamina did not reveal overt synaptic structural changes, suggesting a more subtle, predominantly functional defect. Since retromer is present throughout embryonic and larval development, we cannot exclude the possibility of developmental defects due to *Vps29* loss-of-function. Indeed, Vps35 was mislocalized in the *Vps29* mutant larval brain and we documented increased numbers of

NMJ synaptic boutons. Nevertheless, the adult brain appeared grossly normal, and electroretinograms and climbing were unaffected in newly eclosed flies.

In sum, we consider two non-exclusive models for potential retromer requirement(s) at the synapse. First, retromer may regulate the trafficking of a key factor required for synaptic membrane endocytosis and/or replenishing of the vesicle pool from synaptic endosomes. Second, *Vps29*-related synaptic dysfunction may instead be related, more broadly, to progressive lysosomal dysfunction. In fact, recent studies have linked synaptic endosomes, lysosomes, and autophagosomes to presynaptic proteostasis, including during vesicle recycling and release (*Jin et al., 2018*; *Wang et al., 2017*). For example, another Rab GTPase activating protein, TBC1D24—Skywalker in *Drosophila*—has been shown to regulate synaptic vesicle recycling and quality control via a sorting endosome at NMJ terminals (*Fernandes et al., 2014*; *Uytterhoeven et al., 2011*). Interestingly, in the *Drosophila* visual system, we found that raising *Vps29* or *Vps35* mutants in the dark rescued retinal degeneration but not synaptic transmission defects, consistent with an uncoupling of retromer requirements in these two spatially distinct processes. Thus, it is possible—if not likely—that synaptic endolysosomal trafficking and function can be regulated independently from that of the neuronal soma.

Additional studies will be required to further dissect and refine our understanding of retromer-dependent synaptic mechanisms. For example, it may be informative to perform neurophysiology studies in mutants for other genes encoding retromer partners (e.g. TBC1D5) or regulators of late endosome-lysosome fusion, such as components of the HOPS complex (*Fernandes et al., 2014*; *Lund et al., 2018*). It will also be important to examine synaptic physiology following retromer loss in the mammalian brain. Retromer components are present at synaptic terminals in mouse primary neuronal culture (*Munsie et al., 2015*; *Vazquez-Sanchez et al., 2018*) and in cortical synaptosome preparations (*Tsika et al., 2014*). However, functional studies to date have largely focused on the post-synaptic membrane, where evidence supports retromer requirements for trafficking of glutamatergic and adrenergic neurotransmitter receptors (*Choy et al., 2014*; *Tian et al., 2015*).

## Retromer and neurodegenerative disease

Retromer dysfunction has been linked to pathogenesis of Parkinson's and Alzheimer's disease (*Small and Petsko, 2015*). In both neurodegenerative disorders, synaptic dysfunction is an early pathologic feature, and lysosomal autophagy is also strongly implicated (*Ihara et al., 2012*; *Wang et al., 2017*). Our findings of age-dependent, progressive synaptic and lysosomal dysfunction in the adult brain of *Drosophila* lacking *Vps29* may therefore be relevant for our understanding of retromer in human diseases. In fact, variants at numerous other genetic loci related to endocytic trafficking and synaptic maintenance have been discovered as risk factors for Alzheimer's disease (*BIN1, PICALM, CD2AP, SORL1*) (*Karch et al., 2014*) and Parkinson's disease (*SNCA, LRRK2, SH3GL2/ EndoA, RAB7L1*) (*Soukup et al., 2018*). Genetic links have also recently emerged between Parkinson's disease and lysosomal storage disorders, comprising a heterogeneous group of recessive disorders arising from defects in lysosomal biogenesis and/or function (*Ysselstein et al., 2019*). These disorders, which can include neurodegenerative features, are characterized by the accumulation of undigested substrates and lysosomal expansion (e.g. glucosylceramide in Gaucher's disease). Importantly, retromer participates in the trafficking of lysosomal hydrolases (*Cui et al., 2019*; *Seaman et al., 1997*), and in the absence of *Vps29*, we documented accumulation of glucosylceramides along with ultrastructural evidence of brain lysosomal pathology similar to lysosomal storage disorders. Based on mouse models, synaptic dysfunction may be an early manifestation in lysosomal storage disorders similar to *Vps29* mutant flies (*Mitter et al., 2003*; *Ohara et al., 2004*; *Sambri et al., 2017*). Aberrant lysosomal morphology has also been described in both Alzheimer's and Parkinson's disease models (*Giasson et al., 2002*; *Stokin et al., 2005*) and in human postmortem brain (*García-Sanz et al., 2018*; *Piras et al., 2016*). Targeting the retromer for therapeutic manipulation in neurodegenerative disease is one promising approach, given the availability of small molecule chaperones that bind Vps29 (*Mecozzi et al., 2014*; *Vagnozzi et al., 2019*; *Young et al., 2018*). However, since the endolysosomal system has many functions, including within neurons and other cell types, it will be essential to pinpoint the specific retromer mechanism(s) responsible for age-related brain diseases.

# Materials and methods

## Key resources table

| Reagent type (species) or resource | Designation | Source or reference | Identifiers | Additional information |
|---|---|---|---|---|
| Gene (*D. melanogaster*) | *Vps35* | | FLYB: FBgn0034708 | |
| Gene (*D. melanogaster*) | *Vps29* | | FLYB: FBgn0031310 | |
| Gene (*D. melanogaster*) | *Vps26* | | FLYB: FBgn0014411 | |
| Gene (*D. melanogaster*) | *TBC1D5* | | FLYB: FBgn0038129 | |
| Gene (*D. melanogaster*) | *Rab7* | | FLYB: FBgn0015795 | |
| Genetic reagent (*D. melanogaster*) | FRT40A | Bloomington *Drosophila* Stock Center | FLYB: FBti000207; RRID:BDSC_1816 | $y^1 w^{1118}$; P{neoFRT}40A |
| Genetic reagent (*D. melanogaster*) | ey-FLP; FRT40A | Bloomington *Drosophila* Stock Center | FLYB: FBti0015982; RRID:BDSC_5622 | $y^{d2}$, $w^{1118}$, ey-FLP,GMR-lacZ; P{neoFRT}40A,w+,cl/CyO,y+ |
| Genetic reagent (*D. melanogaster*) | $ey^{3.5}$-FLP; FRT40A | PMID:15848801 | | y w $ey^{3.5}$-FLP; FRT40A,cl, w+/CyO, Kr > GFP |
| Genetic reagent (*D. melanogaster*) | nSyb-Gal4 | Bloomington *Drosophila* Stock Center | FLYB: FBti0150361; RRID:BDSC_51635 | $y^1 w^*$; P{nSyb-GAL4.S}3 |
| Genetic reagent (*D. melanogaster*) | nos-Cas9 | Bloomington *Drosophila* Stock Center | FLYB: FBti0159183; RRID:BDSC_54591 | $y^1$, w*,M{nos-Cas9.P}ZH-2A |
| Genetic reagent (*D. melanogaster*) | $Vps35^{MH20}$/CyO | Bloomington *Drosophila* Stock Center | FLYB: FBal0221635; RRID:BDSC_67202 | w; P{neoFRT}42D, $Vps35^{MH20}$/CyO, Kr > GFP |
| Genetic reagent (*D. melanogaster*) | Df | Bloomington *Drosophila* Stock Center | FLYB: FBti0073468; RRID:BDSC_7491 | Df(2L)Exel6004 |
| Genetic reagent (*D. melanogaster*) | UAS-deGradFP | Bloomington *Drosophila* Stock Center | FLYB: FBti0147362; RRID:BDSC_38421 | w*; P{w[+mC]=UAS-Nslmb-vhhGFP4}3 |
| Genetic reagent (*D. melanogaster*) | GMR-w-RNAi | Bloomington *Drosophila* Stock Center | FLYB: FBti0074622; RRID:BDSC_32067 | GMR-w-RNAi[13D] |
| Genetic reagent (*D. melanogaster*) | UAS-Spin-GFP | Bloomington *Drosophila* Stock Center | FLYB:FBti0148827 RRID:BDSC_39668 | w*; P{UAS-spin.myc-EGFP} |
| Genetic reagent (*D. melanogaster*) | $Vps35^{GFP}$ | This study | | Progenitor = w; $Vps35^{TagRFP-T}$; TagRFP-T cassette was replaced by EGFP |
| Genetic reagent (*D. melanogaster*) | $Vps35^{RFP}$ | PMID:26700726 | | w; $Vps35^{TagRFP-T}$ |
| Genetic reagent (*D. melanogaster*) | eyFLP;FRT42D | PMID:24781186 | | y w,eyFLP,GMR-lacZ; P{neoFRT}42D,w+,cl/CyO,Kr > GFP |
| Genetic reagent (*D. melanogaster*) | UAS-Vps35 | PMID:24781186 | | y w; PBac{UAS-Vps35-HA} |
| Genetic reagent (*D. melanogaster*) | $Rab7^{EYFP}$ | Bloomington *Drosophila* Stock Center | FLYB: FBst0062545 RRID:BDSC_62545 | $w^{1118}$; TI{TI}$Rab7^{EYFP}$ |
| Genetic reagent (*D. melanogaster*) | $Rab7^{Gal4-KO}$/TM3,Sb | PMID:24327558 | FLYB: FBal0294205 | FRT82B, $Rab7^{Gal4-KO}$/TM3,Sb |
| Genetic reagent (*D. melanogaster*) | UAS-Venus-$Rab7^{Q67L}$/CyO | PMID:24327558 | FLYB: FBal0294206 | PBac{UAS-Rab7.Q67L.Venus} |
| Genetic reagent (*D. melanogaster*) | UAS-Venus-$Rab7^{T22N}$/CyO | PMID:24327558 | FLYB: FBal0294207 | PBac{UAS-Rab7.T22N.Venus} |
| Genetic reagent (*D. melanogaster*) | UAS-Venus-$Rab7^{WT}$/CyO | PMID:24327558 | FLYB: FBal0294208 | PBac{UAS-Rab7.WT.Venus} |

*Continued on next page*

*Continued*

| Reagent type (species) or resource | Designation | Source or reference | Identifiers | Additional information |
|---|---|---|---|---|
| Genetic reagent (*D. melanogaster*) | *Vps29[1]/ CyO,twi-GFP* | This study | | fly strain carrying the *ywing[2+]* dominant marker replacing the gene *Vps29* |
| Genetic reagent (*D. melanogaster*) | *Vps29[WT.GFP]/ CyO,ubi-GFP* | This study | | fly strain carrying the *Vps29* gene with EGFP...(GGS)4 sequence inserted at the N-terminus of Vps29 |
| Genetic reagent (*D. melanogaster*) | *Vps29[L152E.GFP]/ CyO,ubi-GFP* | This study | | fly strain carrying the *Vps29* gene with EGFP... (GGS)4 sequence inserted at the N-terminus of Vps29. L152 amino acid is mutated to E |
| Genetic reagent (*D. melanogaster*) | *Vps29-GR* | This study | | CH322-128G03 (FLYB: FBcl0761727) |
| Genetic reagent (*D. melanogaster*) | *UAS-dVps29* | This study | | *y w; PBac{UAS-dVps29-myc}* |
| Genetic reagent (*D. melanogaster*) | *UAS-hVps29-1* | This study | | *y w; PBac{UAS-hVps29-1-FLAG}* |
| Genetic reagent (*D. melanogaster*) | *UAS-hVps29-2* | This study | | *y w; PBac{UAS-hVps29-2-FLAG}* |
| Genetic reagent (*D. melanogaster*) | *UAS-hVps29-3* | This study | | *y w; PBac{UAS-hVps29-3-FLAG}* |
| Genetic reagent (*D. melanogaster*) | *UAS-dTBC1D5* | This study | | *y w; PBac{UAS-dTBC1D5-myc}* |
| Genetic reagent (*D. melanogaster*) | *w; FRT40A, Vps29[1]/CyO* | This study | | Recombinant between *FRT40A* and *Vps29[1]* |
| Genetic reagent (*D. melanogaster*) | *w;Vps29[1],Vps35[GFP]/CyO* | This study | | Recombinant between *Vps35[GFP]* and *Vps29[1]* |
| Genetic reagent (*D. melanogaster*) | *w;Vps29[1]/CyO; Vps29-GR/TM6B* | This study | | Derived by crosses |
| Genetic reagent (*D. melanogaster*) | *w;Vps29[1]/CyO; Rab7[EYFP]/TM6B* | This study | | Derived by crosses |
| Genetic reagent (*D. melanogaster*) | *w;Vps29[1]/CyO; nSyb-Gal4/TM6B* | This study | | Derived by crosses |
| Genetic reagent (*D. melanogaster*) | *w;Vps29[1]/CyO; UAS-dVps29/TM6B* | This study | | Derived by crosses |
| Genetic reagent (*D. melanogaster*) | *w;Vps29[1]/CyO; UAS-hVps29/TM6B* | This study | | Derived by crosses |
| Genetic reagent (*D. melanogaster*) | *w;Vps29[1]/CyO; UAS-dTBC1D5/TM6B* | This study | | Derived by crosses |
| Genetic reagent (*D. melanogaster*) | *w;Vps29[1]/CyO; UAS-dVps35/TM6B* | This study | | Derived by crosses |
| Antibody | FITC-conjugated anti-GFP (mouse monoclonal) | Santa Cruz Biotechnology | Cat# sc-9996 FITC; RRID:AB_627695 | 1:100 for IF |
| Antibody | Anti-Rab7 (mouse monoclonal) | Developmental Studies Hybridoma Bank | RRID:AB_2722471 | 1:100 for IF; 1:1000 for WB |
| Antibody | Anti-Elav (Rat monoclonal) | Developmental Studies Hybridoma Bank | DSHB: Elav-7E8A10; RRID:AB_528218 | 1:500 for IF |
| Antibody | Anti-tubulin Clone DM1A (mouse monoclonal) | Sigma Aldrich | Cat# T6199; RRID:AB_477583 | 1:1000 for WB |

*Continued on next page*

*Continued*

| Reagent type (species) or resource | Designation | Source or reference | Identifiers | Additional information |
|---|---|---|---|---|
| Antibody | Anti-Vps29 (goat polyclonal) | LifeSpan Biosciences | Cat# LS-C55674; RRID:AB_2214913 | 1:2000 for WB |
| Antibody | Anti-Vps35 (Guinea pig polyclonal) | This study | | against C-terminal 338 amino acids of fly Vps35; 1:2000 for WB |
| Antibody | Anti-Vps26 (Guinea pig polyclonal) | PMID:24781186 | | 1:2000 for WB 1:500 for IF |
| Antibody | Anti-CTSL (mouse monoclonal) | R and D Systems | Cat# MAB22591; RRID:AB_2087830 | 1:1000 for WB |
| Antibody | Anti-CTSD (goat polyclonal) | Santa Cruz Biotechnology | Cat# sc-6487; RRID:AB_637895 | 1: 500 for WB |
| Antibody | Rabbit anti-Atg8 | PMID:27068460 | | 1:1000 for WB |
| Antibody | Anti-Polyubiquitinylated conjugates Clone FK1 (mouse monoclonal) | Enzo Life Sciences | Cat# BML-PW8805-0500; RRID:AB_2052280 | 1:1000 for WB |
| Antibody | Rabbit anti-p62/Ref(2)p | PMID:25686248 | | 1:2000 for WB |
| Antibody | Rabbit anti-Arl8 | Developmental Studies Hybridoma Bank | DSHB Cat# Arl8, RRID:AB_2618258 | 1:1000 for IF |
| Antibody | Anti-actin clone C4 (mouse monoclonal) | Millipore | Cat# MAB1501; RRID:AB_2223041 | 1:1000 for WB |
| Antibody | Cy3-conjugated anti-HRP | Jackson ImmunoResearch | Cat# 123-165-021; RRID:AB_2338959 | 1:150 for IF |
| Antibody | Cy3-conjugated Goat anti-mouse IgG | Jackson ImmunoResearch | Cat# 115-165-146; RRID:AB_2338690 | 1:500 for IF |
| Antibody | Cy3-conjugated Goat anti-Rat IgG | Jackson ImmunoResearch | Cat# 112-165-003; RRID:AB_2338240 | 1:500 for IF |
| Antibody | Mouse anti-goat IgG-HRP | Santa Cruz Biotechnology | Cat# sc-2354, RRID:AB_628490 | 1:5000 for WB |
| Antibody | Goat anti-mouse IgG-HRP | Santa Cruz Biotechnology | Cat# sc-2005; RRID:AB_631736 | 1:5000 for WB |
| Antibody | Goat anti-rabbit IgG-HRP | Santa Cruz Biotechnology | Cat# sc-2004; RRID:AB_631746 | 1:5000 for WB |
| Antibody | Rabbit anti-Glc-Cer | Glycobiotech | Cat# RAS_0010 | 1:250 for IF |
| Recombinant DNA reagent | Plasmid: pUASTattb_dVps29-myc | This study | | Progenitors: GH25884 (cDNA) |
| Recombinant DNA reagent | Plasmid: pUASTattb_dTBC1D5-myc | This study | | Progenitors: BS16827 (cDNA) |
| Recombinant DNA reagent | Plasmid: pUASTattb_hVps29-1-FLAG | This study | | Progenitors: OHu00442D (cDNA) |
| Recombinant DNA reagent | Plasmid: pUASTattb_hVps29-2-FLAG | This study | | Progenitors: OHu02289D (cDNA) |
| Recombinant DNA reagent | Plasmid: pUASTattb_hVps29-3-FLAG | This study | | Progenitors: OHu05688D (cDNA) |

*Continued on next page*

*Continued*

| Reagent type (species) or resource | Designation | Source or reference | Identifiers | Additional information |
|---|---|---|---|---|
| Commercial assay or kit | Subcloning Efficiency DH5α competent cells | Thermo Fisher Scientific | Cat# 18265017 | |
| Commercial assay or kit | Q5 Site-Directed Mutagenesis Kit | NEB | Cat# E0554S | |
| Commercial assay or kit | NEBuilder HiFi DNA Assembly Cloning Kit | NEB | Cat# E5520S | |
| Commercial assay or kit | PureLink Genomic DNA Kits | Thermo Fisher Scientific | Cat# K182001 | |
| Commercial assay or kit | GFP-Trap agarose beads | Allele Biotechnology | Cat# ABP-NAB-GFPA100 | |
| Chemical compound, drug | 2X Laemmli Sample Buffer | Bio-Rad | Cat# 161–0737 | |
| Chemical compound, drug | 4% paraformaldehyde in 1XPBS | ChemCruz | Cat# sc-281692 | |
| Chemical compound, drug | RapiClear | SunJin Lab Co. | | |
| Chemical compound, drug | Vectashield | Vector Laboratories | Cat# H-1000 | |
| Chemical compound, drug | Protein A/G agarose | Thermo Fisher | Cat# 20421 | |
| Chemical compound, drug | 8% glutaraldehyde | EMS | Cat# 16020 | |
| Chemical compound, drug | Cacodylic Acid, Trihydrate Sodium 100 g | EMS | Cat# 12300 | |
| Chemical compound, drug | EM-grade glutaraldehyde, 25% Aq solution | EMS | Cat# 16221 | |
| Chemical compound, drug | Osmium tetroxide 4% Aq solution | EMS | Cat# 19191 | |
| Chemical compound, drug | Paraformaldehyde 16% Aq Solution | EMS | Cat# 15711 | |
| Chemical compound, drug | Propylene Oxide | EMS | Cat# 20411 | |
| Chemical compound, drug | Koptec 200 Proof 100% ethanol Anhydrous | VWR | Cat# 89125–186 | |
| Chemical compound, drug | Embed-812 | EMS | Cat# 14901 | |
| Chemical compound, drug | NMA | EMS | Cat# 19001 | |
| Chemical compound, drug | DDSA | EMS | Cat# 13711 | |
| Chemical compound, drug | DMP-30 | EMS | Cat# 13600 | |
| Chemical compound, drug | Uranyl Acetate | EMS | Cat# RT22400 | |
| Chemical compound, drug | Lead Nitrate | EMS | Cat# RT17900-25 | |

*Continued on next page*

*Continued*

| Reagent type (species) or resource | Designation | Source or reference | Identifiers | Additional information |
|---|---|---|---|---|
| Chemical compound, drug | Phalloidin 488 nm | ThermoFisher | Cat# AB_2315147 | |
| Chemical compound, drug | FM 1-43FX | Invitrogen | Cat# F35355 | |
| Chemical compound, drug | Western Lightning Plus-ECL | PerkinElmer | Cat# 121001EA | |
| Software, algorithm | Leica Application Suite X | Leica | RRID:SCR_013673 | |
| Software, algorithm | LabChart Reader | ADInstruments | https://www.adinstruments.com/products/labchart-reader | |
| Software, algorithm | ImageJ | National Institute of Health | RRID:SCR_003073 | |
| Sequence-based reagent | dVps29-myc-F | This study | | 5'-GAAGATCTTCATGCTCGTTCTGGTACTCGGCGA-3' |
| Sequence-based reagent | dVps29-myc-R | This study | | 5'-GCTCTAGACTACAGATCCTCTTCTGAGATGAGTTTTTGTTCGATCTTCTTGTACTCGATGCGCTCCA-3' |
| Sequence-based reagent | dTBC1D5-myc-F | This study | | 5'-GAAGATCTATCAACATGACTGTTTGGGGAATAGAAGCCATCA-3' |
| Sequence-based reagent | dTBC1D5-myc- R | This study | | 5'-GCTCTAGATCACAGATCCTCTTCTGAGATGAGTTTTTGTTCACTCGATTCGTTTCGATGCCGT-3' |
| Sequence-based reagent | hVps29-1-F | This study | | 5'-CCGCCTCGAGGCCACCATGTTGGTGTTGGT-3' |
| Sequence-based reagent | hVps29-2-F | This study | | 5'-CCGCCTCGAGGCCACCATGGCTGGGCACA-3' |
| Sequence-based reagent | hVps29-3-F | This study | | 5'-CCGCCTCGAGGCCACCATGAGCAGGTGTGCT-3' |
| Sequence-based reagent | hVps29-R | This study | | 5'-CTAGTCTAGATTATCACTTATCGTCGTCATCCTTGTAATCAGGT-3' |
| Sequence-based reagent | Vps29-P1-F | This study | | 5'-GAACCTGACGTATCCGGAGC-3' |
| Sequence-based reagent | Vps29-P1-R | This study | | 5'-TCGCCGATCAGTTGGTACAC-3' |
| Sequence-based reagent | Vps29-P2-F | This study | | 5'-CTCGTTCTGGTACTCGGCG-3' |
| Sequence-based reagent | Vps29-P2-R | This study | | 5'-ACGAACGAAGGCACCACATT-3' |
| Sequence-based reagent | Vps29-P3-F | This study | | 5'-GGCCGCATACATCACATCCT-3' |
| Sequence-based reagent | Vps29-P3-R | This study | | 5'-GAATTTGTTGCCGTGCTCGT-3' |
| Sequence-based reagent | Vps35-F (Internal control) | This study | | 5'-TTGTACCTCCTCATAACAGTGGG-3' |
| Sequence-based reagent | Vps35-R (Internal control) | This study | | 5'-TCGTTCTCCTCAACCATCACAT-3' |
| Other | Leica SP8 confocal microscope | Leica | | |
| Other | Leica DM 6000 B system | Leica | | |
| Other | Zeiss LSM 880 with Airyscan | Zeiss | | |

## Fly stocks and husbandry

A complete list of fly strains used in this study is included in the Key Resources Table. Detailed information on experimental genotypes can also be found in figures and figure legends. The $Vps29^1$ null allele was generated using CRISPR/Cas9-mediated gene replacement, as previously described (*Li-Kroeger et al., 2018*). Briefly, a dominant marker $ywing^{2+}$ flanked by homology arms on each side, along with sgRNA expression plasmids, were injected into embryos expressing Cas9 ($y^1,w*,M\{nos-Cas9.P\}ZH-2A$). G0 animals were next crossed to $y\ w$, and progeny were screened for the presence of the $yellow^+$ wing marker to identify $Vps29^1$ ($Vps29^{ywing2+}$). Studies of $Vps29$ mutant flies used either $Vps29^1$ homozygotes or the transheterozygous genotype, $Vps29^1/Df(2L)Exel6004$ (referred to as $Vps29^1/Df$). To generate the $Vps29$ genomic rescue strain, a 23 kb P[acman] genomic fragment (CH322-128G03) including the $Vps29$ locus was injected into $y,w,\Phi C31;\ VK33\ attP$ embryos, followed by selection of F1 transformants. All studies using the transgenic $Vps29$ BAC examined the genomic rescue construct ($GR$) in heterozygosity (either $Vps29^1;GR/+$ or $Vps29^1/Df;GR/+$).

In order to generate the $Vps29^{WT.GFP}$ and $Vps29^{L152E.GFP}$ alleles, we again used the CRISPR/Cas9 gene-replacement system, starting with the $Vps29^1$ strain. The GFP coding sequence was cloned in-frame and proximal to the $Vps29$ cDNA using NEBuilder HiFi DNA Assembly Cloning Kit (NEB). Using Q5 Site-Directed Mutagenesis Kit (NEB), the L152E mutation was introduced to the $GFP::Vps29$ plasmid. As above, the $GFP::Vps29$ or $GFP::Vps29^{L152E}$ constructs were each injected into $y^1\ M\{nos-Cas9.P\}ZH-2A\ w*;\ Vps29^1/CyO$ flies. G0 flies were crossed to $y\ w$, and F1 progeny were screened for loss of the $yellow^+$ marker to establish the $Vps29^{WT.GFP}$ and $Vps29^{L152E.GFP}$ strains. The sequence of all CRISPR/Cas9-generated strains were confirmed by genomic PCR followed by Sanger sequencing. Genomic DNA from a single fly was prepared using 'squish buffer' (10 mM Tris-Cl pH 8.0, 1 mM EDTA, 25 mM NaCl, and 0.2 mg/ml Proteinase K). Total genomic DNA was extracted using the PureLink Genomic DNA Kit (Thermo Fisher Scientific). PCR primers used for additional confirmation of the $Vps29^1$ allele (*Figure 1B*) are listed in the Key Resources Table.

As part of this study, we also generated transgenic strains carrying $UAS-dVps29-myc$, $UAS-dTBC1D5-myc$, and $UAS-hVps29-FLAG$. The full-length *Drosophila* Vps29 cDNA (*Drosophila* Genomics Resource Center (DGRC) plasmid: GH25884) was cloned proximal to $c-myc$ coding sequencing, using primers $dVps29-myc-F$ and $dVps29-myc-R$ (Key Resources Table). The full-length cDNA for $TBC1D5$ (DGRC plasmid: BS16827) was similarly cloned proximal to the $c-myc$ coding sequencing using primers $dTBC1D5-myc-F$ and $dTBC1D5-myc-R$. The $Vps29-myc$ and $TBC1D5-myc$ PCR products were each digested with BglII and XbaI restriction enzymes (NEB) and ligated to the $pUAST-attB$ vector using T4 ligase (NEB). Plasmids containing FLAG-tagged Human VPS29 cDNAs (OHu00442D, OHu02289D, and OHu05688D, respectively) were obtained from GenScript. For PCR cloning, we used $hVps29-R$ along with each of the following: $hVps29-1-F$ for hVps29 transcript variant 1 (NM_016226.4), $hVps29-2-F$ for hVps29 transcript variant 2 (NM_057180.2), and $hVps29-3-F$ for hVps29 transcript variant 3 (NM_001282150.1). PCR products were digested by XhoI and XbaI restriction enzymes (NEB), and then ligated to the $pUAST-attB$ vector. The resulting constructs were purified, verified by Sanger DNA sequencing, and injected into $y,w,\Phi C31;\ VK33$ embryos to generate transgenic flies. The generation of $w;Vps35^{TagRFP-T}$ ($Vps35^{RFP}$) with a Rippase-switchable TagRFP-T to GFP tag was described in *Koles et al. (2016)*. $w;Vps35^{GFP}$ was generated by crossing $Vps35^{RFP}$ to $bam-Gal4;;UAS-Rippase::PEST$ @$attP2$ flies to isolate germline ripouts of the TagRFP-T cassette, leaving a C-terminal EGFP knockin.

All *Drosophila* crosses were raised on molasses-based food at 25°C unless otherwise noted. To induce neuronal specific knockdown of Vps35$^{GFP}$, we used the deGradFP system (*Caussinus et al., 2012*; *Nagarkar-Jaiswal et al., 2015*). Crosses were established and maintained at 18°C, and following eclosure, adults were shifted to 29°C for 7 days before brain dissection.

## Histology and immunofluorescence

For histology, *Drosophila* heads were fixed in 8% glutaraldehyde (Electron Microscopy Sciences) at 4°C for 6 days and embedded in paraffin. Frontal (5 μm) sections were cut using a Leica Microtome (RM2245), mounted on slides, and stained with hematoxylin and eosin. For whole mount brain immunostaining, adult fly brains were dissected in 1X PBS and fixed at 4°C overnight in 4% paraformaldehyde in 1X PBS (sc-281692, ChemCruz). Fixed brains were permeabilized with 2% PBST (1X PBS+2% Triton X-100) at 4°C for 20 hr, and then vacuumed (Nalgene) at room temperature for 1 hr. Brains

were incubated with the primary antibodies at 4°C for 2 days, followed by secondary antibody incubation for 1 day at 4°C, then mounted in RapiClear (SunJin Lab Co.). For NMJ staining, wandering third instar larvae were dissected in Hemolymph-Like 3 (HL3) solution without calcium (110 mM NaCl, 5 mM KCl, 10 mM NaHCO$_3$, 10 mM MgCl$_2$, 5 mM trehalose, 30 mM sucrose, and 5 mM HEPES, pH 7.2), fixed in 100% methanol at room temperature for 20 min, washed in 0.3% PBST, and blocked for 1 hr in 0.3% PBST containing 5% normal goat serum. Tissues were incubated with primary antibody overnight at 4°C, followed by secondary antibodies at room temperature for 2 hr, and mounted using Vectashield (Vector Laboratories) prior to imaging. All NMJ studies focused on muscles 6/7, abdominal section 2 and 3. To quantify GFP (Vps35) intensity within NMJ synaptic boutons, confocal regions of interest (ROIs) were determined using the Imaris (Bitplane) 'surface' tool to select HRP-positive regions. Identical parameters for defining HRP surfaces were applied for all samples. Mean GFP signal intensity within the ROI encompassing the synaptic area was normalized to HRP volume (μm$^3$). For fly retina staining, fly heads were fixed after the removal of proboscis in 3.7% formaldehyde in PBS at 4°C overnight. Fixed retinae were dissected and blocked in 0.1% PBST containing 5% natural donkey serum at room temperature for 1 hr, followed by primary antibody incubation for 2 days at 4°C and secondary antibody incubation for 2 hr at room temperature. Primary antibodies were used at following dilutions in 0.3% PBST: FITC-conjugated anti-GFP (sc-9996 FITC, 1:100, Santa Cruz Biotechnology, RRID:AB_627695), Mouse anti-Rab7 (1:100, DSHB, RRID:AB_2722471), Rat anti-Elav (7E8A10,1:500, DSHB, RRID:AB_528218), Rabbit anti-GlcCer (1:250, Glycobiotech); Cy3-conjugated anti-HRP (1:150, Jackson ImmunoResearch); Rabbit anti-Arl8 (1:1000, DSHB, RRID:AB_2618258); Guinea pig anti-Vps26 (1:500, *Wang et al., 2014*). For secondary antibodies, we used Cy3-conjugated goat anti-mouse or anti-Rat IgG, and Alexa 647-conjugated goat anti-Rabbit or anti-Guinea pig IgG (1:500, Jackson ImmunoResearch). Confocal microscopy images were acquired with a Model LSM 880 confocal system (Carl Zeiss).

## Western blot analysis and immunoprecipitation

For western blots, adult fly heads or whole larvae were homogenized in 2X Laemmli Sample Buffer (Bio-Rad) with 5% β-mercaptoethanol (Calbiochem) using a pestle (Argos Technologies). The lysates were heated at 95°C for 5 min, followed by centrifugation at 21,130 × g at 4°C for 15 min before SDS-PAGE analysis. Samples were loaded into 12% Bis-Tris gels (Invitrogen), separated by SDS-PAGE. For immunoprecipitation of Vps35$^{GFP}$, 300 fly heads from 1- to 2 day-old animals were homogenized on ice in 700 μL of lysis buffer containing 10 mM Tris pH 7.5, 150 mM NaCl, 0.5 mM EDTA, 0.5% NP-40, and 1X complete protease inhibitor (Roche). Homogenized samples were centrifugated at 21,130 × g for 30 min at 4°C. 3% (by volume) of the supernatant were reserved for total protein input. The remaining supernatant was incubated for 1 hr at 4°C with 50 μL of protein A/G agarose slurry (Thermo Fisher) to reduce non-specific binding. Following pre-clearing, the lysate was incubated for 3 hr at 4°C with 35 μL of GFP-Trap agarose beads (Allele Biotechnology) with mild agitation. The beads were washed three times with lysis buffer, boiled in 70 μL of 2X Laemmli sample buffer (Biorad), and subjected to SDS-PAGE (BOLT12% Bis-Tris Gel, Invitrogen). Gels were transferred to PVDF membrane (Millipore), and blocked in 5% bovine serum albumin (Sigma) in 1X TBST (Tris-buffered saline + 0.1% Tween-20). We used the following primary antibodies and dilutions: mouse anti-tubulin (DM1A, Sigma Aldrich, RRID:AB_477583), 1:1000; mouse anti-Rab7 (DSHB, RRID:AB_2722471), 1:1000; Goat anti-Vps29 (LS-C55674, LifeSpan Biosciences, RRID:AB_2214913),1:2000; Guinea pig anti-Vps26 (*Wang et al., 2014*), 1:2000; Guinea pig anti-Vps35 (see below), 1:2000; Mouse anti-CTSL (clone 193702, MAB22591, R and D Systems, RRID:AB_2087830), 1:2000; Goat anti-CTSD (sc-6487, Santa Cruz, RRID:AB_637895): 1:500, Rabbit anti-p62/Ref(2)p (*Rui et al., 2015*), 1:2000; Rabbit anti-Atg8 (*Castillo-Quan et al., 2016*), 1:1000; Mouse anti-GFP (B-2, Santa Cruz, RRID:AB_627695), 1:1000; Mouse anti-actin (C4, Millipore, RRID:AB_2223041), 1:1000; Mouse anti-Polyubiquitinylated proteins (clone FK1, Enzo, RRID:AB_2052280), 1:1000. HRP-conjugated secondary antibodies were used at 1:5000. Antibodies against *Drosophila* Vps35 were generated by immunization of guinea pigs with a bacterially expressed peptide comprising the C-terminal 338 amino acids of Vps35. Bacterial expression as a GST fusion protein was facilitated by the pGEX 6 P-2 vector. The GST tag was used for protein purification and was cleaved off using PreScission Protease (GE Healthcare Life Sciences) prior to immunization.

## Survival and climbing assays

For survival analyses, around 200 flies per genotype (approximately equal numbers of females and males) were aged in groups of no more than 30 flies per vial, transferring to fresh vials and food every 2–3 days. For the startle-induced negative geotaxis assay (climbing), four to five groups consisting of approximately 15 flies each were placed in a plastic cylinder. Flies were gently tapped to the bottom of the cylinder, and locomotor activity was videotaped and quantified as the percentage of animals climbing past the 9 cm line within a 15 s interval.

## Electroretinogram (ERG)

All crosses were initially maintained in the dark to prevent flies from being exposed to light before eclosion. Newly-eclosed adults were either shifted to a 12 hr light/dark cycle (3,000 lux for light exposure) or maintained in light-sealed boxes for constant darkness. During aging, the positions of light-exposed vials were randomly shuffled within racks. ERG recordings were performed as previously described (*Chouhan et al., 2016*). In brief, adult flies were anesthetized and glued to a glass slide, with electrodes placed on the corneal surface and the thorax. Flies were maintained in the dark for at least 1 min prior to stimulation using a train of alternating light/dark pulses. Retinal responses were recorded and analyzed using LabChart software (ADInstruments). At least eight flies were examined for each genotype and timepoint.

## NMJ electrophysiology and FM 1–43 dye studies

Animals were dissected at the wandering third instar stage in HL3 buffer (above), without addition of calcium. Electrophysiologic recordings were performed with 0.5 mM extracellular $Ca^{2+}$ buffer concentrations, as described in *Ojelade et al. (2019)*. Larval motor axons were severed and miniature excitatory junction potentials (mEJPs) were recorded from muscle 6 of abdominal segments A2 and A3 for 2 min. EJPs were evoked at 0.2 Hz. EJPs and mEJPs were analyzed using pClamp6 (Molecular Devices) and Mini Analysis Program (Synaptosoft) software, respectively. High-frequency stimulation was recorded at 10 Hz for 10 min. The recorded EJP data were binned at 0.5 min intervals and normalized to the average EJP. EJP amplitudes were corrected for nonlinear summation as previously described (*Martin, 1955*). The FM 1–43 dye uptake assay was performed as described in *Verstreken et al. (2008)*. Larval NMJ preparations were stimulated for 2 min in dark conditions using HL3 buffer containing 90 mM $K^+$, 1.5 mM $Ca^{2+}$, and 4 µM FM 1–43 dye. Larval preparations were subsequently washed five times with HL3 solution (without 90 mM $K^+$ and calcium), and FM dye uptake was imaged using a Leica Sp8 confocal system. Signal intensity of FM dye per bouton was normalized to each bouton area. Six boutons were sampled per animal. For each genotype, 7–11 animals were assayed.

## Transmission Electron Microscopy (TEM)

For analysis of adult eye ultrastructure, fly heads were dissected by removing proboscis and air sacs, then severed from the thorax. For analysis of central brain ultrastructure, fly heads were dissected by removing cuticles and eyes to ensure complete penetration of fixative, leaving each head affixed to the thorax. All samples were processed for TEM as previously described (*Chouhan et al., 2016*) using a Ted Pella Bio Wave processing microwave with vacuum attachment. Briefly, dissected samples were fixed (4% paraformaldehyde, 2% glutaraldehyde, and 0.1 M sodium cacodylate, pH 7.2) at 4°C for 48 hr, and then submerged in 1% osmium tetroxide for 45 min. The fixed samples were dehydrated using an ethanol series followed by propylene oxide, and then embedded using Embed-812 resin (EMS). For brain samples, each head was removed carefully from the thorax prior to embedding. Transverse sections of brains (50 nm) and tangential sections of eyes (50 nm) were prepared with a Leica UC7 microtome, and post-stained with 1% uranyl acetate and 2.5% lead citrate. All TEM images were acquired using a JEOL 1010 Transmission Electron Microscope. TEM images of photoreceptor sections were prepared from three different animals per genotype. TEM images of brain sections were prepared from four animals per genotype. Images were acquired from the dorsal-posterior cortical brain region (*Figure 7—figure supplement 1D*), including at least 50 cells per brain.

## Quantification and statistics

Confocal images were processed and analyzed using ImageJ software (NIH). Sample size for all comparisons is included in each figure legend (also noted above). For survival curves, we performed log-rank test with Bonferroni's correction. For other statistical analysis, we performed two-tailed, unpaired t-tests or Analysis of Variance (ANOVA) followed by Tukey's post-hoc test for multiple comparisons, as specified in all figure legends. The significance threshold for all analyses was set to $p < 0.05$. Otherwise, results are noted as 'not significant' (n.s.). Error bars in all analyses represent the standard error of the mean (SEM).

## Acknowledgements

We thank Drs. L Partridge, S Zhang, and PR Hiesinger for providing antibodies and *Drosophila* stocks. We also thank the Bloomington *Drosophila* Stock Center, the Vienna *Drosophila* RNAi Center, the Developmental Studies Hybridoma Bank, and FlyBase (*Gramates et al., 2017*). We thank Y He and L Duraine for research technical support. We are grateful to Drs. M Sardiello, K Venkatachalam, J Botas, and H Zoghbi for feedback and discussions.

## Additional information

### Competing interests

Hugo J Bellen: Reviewing editor, *eLife*. The other authors declare that no competing interests exist.

### Funding

| Funder | Grant reference number | Author |
| --- | --- | --- |
| National Institutes of Health | R01AG053960 | Joshua M Shulman |
| National Institutes of Health | U01AG046161 | Joshua M Shulman |
| National Institutes of Health | U01AG061357 | Joshua M Shulman |
| National Institutes of Health | U54AG065187 | Joshua M Shulman |
| Burroughs Wellcome Fund | Career Award for Medical Scientists | Joshua M Shulman |
| National Institutes of Health | R01NS103967 | Avital Adah Rodal |
| National Institutes of Health | U54HD083092 | Joshua M Shulman |
| National Institutes of Health | P30CA125123 | Joshua M Shulman |
| Howard Hughes Medical Institute | | Hugo J Bellen |
| Robert and Renee Belfer Family Foundation | | David Li-Kroeger<br>Hugo J Bellen |
| Alzheimer's Association | AARFD-16-442630 | Shamsideen A Ojelade |
| Burroughs Wellcome Fund | Postdoctoral Enrichment Program Award (BWF-1017399) | Shamsideen A Ojelade |

The funders had no role in study design, data collection and interpretation, or the decision to submit the work for publication.

### Author contributions

Hui Ye, Conceptualization, Formal analysis, Investigation; Shamsideen A Ojelade, David Li-Kroeger, Zhongyuan Zuo, Liping Wang, Yarong Li, Investigation; Jessica YJ Gu, Ulrich Tepass, Avital Adah Rodal, Resources; Hugo J Bellen, Supervision; Joshua M Shulman, Conceptualization, Supervision, Funding acquisition

**Author ORCIDs**

Hui Ye (iD) https://orcid.org/0000-0003-3965-9702
David Li-Kroeger (iD) http://orcid.org/0000-0001-6473-7691
Avital Adah Rodal (iD) http://orcid.org/0000-0002-2051-8304
Hugo J Bellen (iD) http://orcid.org/0000-0001-5992-5989
Joshua M Shulman (iD) https://orcid.org/0000-0002-1835-1971

**Decision letter and Author response**

Decision letter https://doi.org/10.7554/eLife.51977.sa1
Author response https://doi.org/10.7554/eLife.51977.sa2

## Additional files

**Supplementary files**

• Transparent reporting form

**Data availability**

All data generated or analysed during this study are included in the manuscript and supporting files.

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
