## [Decision Letter]

Thank you for submitting your article "Retromer subunit, VPS29, regulates synaptic transmission and is required for endolysosomal function in the aging brain" for consideration by *eLife*. Your article has been reviewed by two peer reviewers, and the evaluation has been overseen by K VijayRaghavan as the Senior and Reviewing Editor. The following individual involved in review of your submission has agreed to reveal their identity: Ole Kjærulff (Reviewer #1).

The reviewers have discussed the reviews with one another and the Reviewing Editor has drafted this decision to help you prepare a revised submission.

Summary:

The manuscript submitted by Ye et al. reports, for the first time, the effect of deleting the *Drosophila* gene encoding a core retromer component, *Vps29*. Intriguingly, in the adult brain this causes a striking redistribution of another core component, Vps35, from the neuropil to somata. Also, in *Vps29* mutants, synaptic endocytosis is compromised at the larval neuromuscular junction. Moreover, with aging the retromer insufficiency associated with *Vps29* loss elicits progressive endolysosomal dysfunction, with ultrastructural evidence of impaired substrate clearance and lysosomal stress.

The manuscript is of high quality. It contains novel and interesting results that should be of broad interest, and the preparation is thorough. Overall, we believe that the results provide a promising starting point for publication in *eLife*. However, we also feel that stronger attempts could be made to substantiate the claims made by the authors and to obtain deeper mechanistic insights.

Essential revisions:

1) In Figure 8 (bottom), the authors propose a molecular model that posits that, in *Vps29* mutants, the two other retromer core components Vps35 and Vps26 should co-localize on endosomes, trapped by Rab7-GTP. Although this model is well in accordance with the evidence obtained from genetic experiments, we feel that more data are required to substantiate it. Specifically, more could be done to characterize the distribution and predicted co-localization of Vps35-GFP and the Rab7 immunosignal at the subcellular (organellar) level, rather than just demonstrating the co-localization at the cellular level in the antennal lobe perinuclei as in the present manuscript. Investigating the subcellular (co-)localization could be done generally, e.g. in the larval CNS or in more "defined" systems, such as the large somata associated with some peptidergic neurons (using, for example, the c929-Gal4 or the CCAP-Gal4 driver).

Another well-suited substrate for these experiments might be the larval garland cells. Although Rab7-GTP and Vps35 in the *Vps29* nulls are likely to accumulate primarily on late endosomes, it would be desirable – and the authors should decide if they can reasonably do this – if this could be verified using independent markers for late endosomes/lysosomes, such as Rab9, LrrK, Spinster, or LAMP1.

2) The authors demonstrate in *Vps29* nulls both that synaptic endocytosis is compromised at the third instar larval NMJ and that Vps35 redistributes/mislocalizes from the neuropil to somata in the adult brain. To explore in more depth – in the same neuronal population, in which synaptic effects were evaluated – the possible mechanistic link between retromer mislocalization and synaptic dysfunction, we are curious to know if, in *Vps29* nulls, Vps35 also mislocalizes in larval motor neurons (Vps35 is stated to be present at the pre-synapse at the NMJ). Would Vps35 redistribute away from the pre-synapse (and possibly from axons) and accumulate in the cell bodies also in these neurons? This experiment should be feasible, e.g. by constructing UAS-Vps35[GFP], *Vps29*[1]; D42-Gal4/+ larvae, where D42-Gal4 is a motor neuron driver located on the third.

3) The authors also point out that the lysosomal function is perturbed in retromer mutants. It is possible that the observed perturbation of endocytosis and synaptic vesicle recycling are indirect effects of lysosomal dysfunction rather than due to a direct requirement for retromer during the earlier stages of the endosomal pathway. The authors could discuss this point and, perhaps, could test this by doing high-frequency stimulation or at least FM1-43 uptake experiments in mutants that abrogate LE-lysosome fusion (such as null mutations for the HOPS complex components Vps39 or Lt/Vps41 – which are viable until early pupal stages). A milder phenotype in mutants that specifically perturb lysosomal function compared to retromer mutants would demonstrate that retromer's effect on endocytosis is directly due to its function in the upstream part of the endosomal pathway.

4) The use of different mutants is not always clear. Sometimes, the mutant over deficiency is used and not the homozygous viable mutant generated by the authors. Is there a specific reason or difference in the phenotypes, e.g. background mutations on the CRISPR chromosome?

The CRISPR null mutant seems to have stronger phenotypes in some cases, like the retinal degeneration phenotype after 20 days shown in Figure 1—figure supplement 1B and or the declined climbing ability of aged flies shown in Figures 3A and B, than the mutant/Df. Especially the retinal degeneration phenotype (see Figure 1—figure supplement 1B) seems to be more pronounced. An experimental comparison of the phenotypes, including rescue, using the same assay (e.g. but not necessarily, TEM) would be useful to assess defects due to *Vps29* independent of the background.

5) Figure 1F and subsection “*Vps29* is required for age-dependent retinal function”, second paragraph: In the depolarization graph, for flies raised 10 days in the dark, the difference between control and null mutant looks comparable to the measured difference at 10 days light-dark stimulation, which is highly significant with three asterisks. What is the precise p-value for the "dark flies"? Is the missing significance here due to a high variance in data points, although this is not shown by the error bar? If the variance is high, why were only 6-8 flies tested here? Here, the single data points should be shown and the n number of flies tested for ERG needs to be increased to at least double the number of flies tested.

Furthermore, the amplitude of the depolarization of null mutant flies raised in dark for 10 days is lower than the amplitude for the LD flies. Although, in the manuscript, this is described as a "rescued phototransduction". Is the amplitude of "dark flies" consistently lower than the LD flies? More flies (again at least double the number of flies tested before) should be tested here to clarify significance.

In general, there is a marked discrepancy for the amplitude between the example data in E and the population data in 1F, both for *Vps29* and Vps35 mutants when looking across the three experimental conditions D1-dark, D1 12h-LD, and D10-dark

6) Figure 3A and B: In A, the percentage of mutant/Df flies passing 9 cm in 15 sec is around 40%. While in B, the percentage of null mutant flies (VPS291; *nsyb-Gal4*) is roughly 10%, although being of the same age as the flies tested in A. How can this significant difference be explained and why weren't the null mutant flies also tested at 2 and 10 days, if they are so different than mutant/Df flies? The authors should test the null mutant flies also for 2 and 10 days to check for any stronger behavior effects compared to the mutant over deficiency flies.

---

## [Author Response]

Essential revisions:1) In Figure 8 (bottom), the authors propose a molecular model that posits that, in Vps29 mutants, the two other retromer core components Vps35 and Vps26 should co-localize on endosomes, trapped by Rab7-GTP. Although this model is well in accordance with the evidence obtained from genetic experiments, we feel that more data are required to substantiate it. Specifically, more could be done to characterize the distribution and predicted co-localization of Vps35-GFP and the Rab7 immunosignal at the subcellular (organellar) level, rather than just demonstrating the co-localization at the cellular level in the antennal lobe perinuclei as in the present manuscript. Investigating the subcellular (co-)localization could be done generally, e.g. in the larval CNS or in more "defined" systems, such as the large somata associated with some peptidergic neurons (using, for example, the c929-Gal4 or the CCAP-Gal4 driver).Another well-suited substrate for these experiments might be the larval garland cells. Although Rab7-GTP and Vps35 in the Vps29 nulls are likely to accumulate primarily on late endosomes, it would be desirable – and the authors should decide if they can reasonably do this – if this could be verified using independent markers for late endosomes/lysosomes, such as Rab9, LrrK, Spinster, or LAMP1.

In order to address these questions and further substantiate our findings, we have performed a number of additional studies, as requested. First, we have performed additional stainings in the adult brain, including 2 independent markers for the late endosome/lysosome, Arl8 and Spinster-GFP. We find that Arl8 is also increased in *Vps29* mutants, and that these proteins show overlapping perinuclear localization in the antennal lobe, along with Vps35 (Figure 4—figure supplement 3A). Rab7 and Arl8 function co-ordinately in late endosome-lysosome fusion (Jongsma et al., 2020; Marwaha et al., 2017). In addition, higher-power imaging of several large neurons in the posterior central brain reveals that the increased Rab7- and Arl8-positive signal assumes a punctate, cytoplasmic distribution that further co-stains with Spinster-GFP (*nsyb-Gal4>UASSpinster-GFP*) (Figure 4—figure supplement 3B). We have added relevant text to the Results (subsection “*Vps29* interacts with Rab7 and *TBC1D5*”, first paragraph) and Discussion (subsection “VPS29 is required for retromer recruitment via Rab7 and *TBC1D5*”, first paragraph). We believe these data further support our model that residual retromer components are trapped with Rab7 at the late endosome/lysosome. As requested, we also examined staining in other regions. In the adult retina of *Vps29* mutants, we found that Vps35, Vps26, and Rab7 are increased in the cortex of the lamina where neuronal cell bodies are concentrated. Lastly, we have also performed additional stainings in larval brain and at the larval NMJ. In the larval brain, we document a striking loss of Vps35-GFP staining in the neuropil regions, and more subtle but consistent reduction of Vps35GFP at synaptic boutons. Although we were not able to appreciate similar redistribution to the neuronal soma in larvae, this may be due to differences in assay sensitivity or mechanisms in the larval vs. adult nervous system. The new data on Vps35 localization is discussed in the Results (subsection “*Vps29* regulates retromer localization and is required in the aging nervous system”, last paragraph) and Discussion (subsection “VPS29 is required for retromer recruitment via Rab7 and *TBC1D5*”, last paragraph), and is shown in Figure 4—figure supplement 1B and Figure 4—figure supplement 2. Overall, our results support an important role for *Vps29* in retromer localization within neurons.

2) The authors demonstrate in Vps29 nulls both that synaptic endocytosis is compromised at the third instar larval NMJ and that Vps35 redistributes/mislocalizes from the neuropil to somata in the adult brain. To explore in more depth – in the same neuronal population, in which synaptic effects were evaluated – the possible mechanistic link between retromer mislocalization and synaptic dysfunction, we are curious to know if, in Vps29 nulls, Vps35 also mislocalizes in larval motor neurons (Vps35 is stated to be present at the pre-synapse at the NMJ). Would Vps35 redistribute away from the pre-synapse (and possibly from axons) and accumulate in the cell bodies also in these neurons? This experiment should be feasible, e.g. by constructing UAS-Vps35[GFP], Vps29[1]; D42-Gal4/+ larvae, where D42-Gal4 is a motor neuron driver located on the third.

As described above, we have examined the localization of Vps35-GFP at the larval NMJ in *Vps29* mutant animals, and we have newly added our results showing a reduction of the synaptic staining (Figure 4—figure supplement 2B and subsection “*Vps29* regulates retromer localization and is required in the aging nervous system”, last paragraph). The striking reduction of Vps35-GFP in the larval brain neuropil (Figure 4—figure supplement 2A) is also consistent, but we do not detect a concomitant enrichment of Vps35 in the motor neuron cell bodies. While it is possible that these incongruities might be resolved with additional studies, such as with tissue specific manipulation with D42-Gal4, we were not able to feasibly undertake such experiments given the time constraints. In our original manuscript, we explicitly separated discussion of the larval NMJ synaptic studies from those performed in the adult brains, given uncertainty as to whether the *Vps29* loss-of-function phenotypes have similar explanations. However, our new data support the possibility that retromer mislocalization indeed contributes to both.

3) The authors also point out that the lysosomal function is perturbed in retromer mutants. It is possible that the observed perturbation of endocytosis and synaptic vesicle recycling are indirect effects of lysosomal dysfunction rather than due to a direct requirement for retromer during the earlier stages of the endosomal pathway. The authors could discuss this point and, perhaps, could test this by doing high-frequency stimulation or at least FM1-43 uptake experiments in mutants that abrogate LE-lysosome fusion (such as null mutations for the HOPS complex components Vps39 or Lt/Vps41 – which are viable until early pupal stages). A milder phenotype in mutants that specifically perturb lysosomal function compared to retromer mutants would demonstrate that retromer's effect on endocytosis is directly due to its function in the upstream part of the endosomal pathway.

We agree with the reviewers that either or both interpretations are consistent with our experimental data. We have attempted to discuss these alternative models at length in the Discussion (subsection “The retromer and endolysosomal trafficking in neurons”), first presenting the evidence in support of primary lysosomal dysfunction followed by the data that supports a synaptic role. We then discuss the 2 alternative mechanistic models “head-to-head”. In our revisions we have made edits to further clarify this discussion, and we have added text to emphasize potential future experimental approaches that might further differentiate between these possibilities. We are eager to pursue this line of investigation, as suggested by reviewers; however, given the time constraints, we were not able to complete and include these experiments as part of this revision.

4) The use of different mutants is not always clear. Sometimes, the mutant over deficiency is used and not the homozygous viable mutant generated by the authors. Is there a specific reason or difference in the phenotypes, e.g. background mutations on the CRISPR chromosome?The CRISPR null mutant seems to have stronger phenotypes in some cases, like the retinal degeneration phenotype after 20 days shown in Figure 1—figure supplement 1B and or the declined climbing ability of aged flies shown in Figures 3A and B, than the mutant/Df. Especially the retinal degeneration phenotype (see Figure 1—figure supplement 1B) seems to be more pronounced. An experimental comparison of the phenotypes, including rescue, using the same assay (e.g. but not necessarily, TEM) would be useful to assess defects due to Vps29 independent of the background.

As noted, we have utilized both the *Vps29^1^*homozygous and *Vps29^1^ / Df* genotypes in our studies, along with a genomic rescue (BAC transgenic) strain (GR) to ensure specificity of all phenotypes studied. As is standard for careful genetic analyses, in several cases, we have examined both genotypes as a further control for potential genetic background effects, including the possibility of unknown linked mutations on the *Vps29^1^*chromosome. In order to address the valid concern raised by the reviewers, we have carefully repeated the locomotor (climbing) assay, including both *Vps29^1^*homozygous and *Vps29^1^ / Df* genotypes, along with key controls (*Vps29^1^; GR/+* and *Vps29^1^ / Df; GR/+*). The new results (Figure 3A) indeed confirm that the *Vps29^1^*homozygote genotype manifests a stronger locomotor dysfunction phenotype (p<0.01 in comparisons at 20 days). Importantly however, both genotypes are fully rescued by the genomic BAC transgene. We conclude that while the *Vps29^1^* chromosome may contain an enhancer (or the Df chromosome, a suppressor), our rescue data confirms the specificity of the climbing phenotype for loss of *Vps29 –* similar to other phenotypes of interest in our study. We have added text to the Results to make this interpretation clearer (subsection “*Vps29* regulates retromer localization and is required in the aging nervous system”, first paragraphs), and our revision also explicitly denotes the genotypes for key experiments to avoid any ambiguity in the manuscript text.

5) Figure 1F and subsection “Vps29 is required for age-dependent retinal function”, second paragraph: In the depolarization graph, for flies raised 10 days in the dark, the difference between control and null mutant looks comparable to the measured difference at 10 days light-dark stimulation, which is highly significant with three asterisks. What is the precise p-value for the "dark flies"? Is the missing significance here due to a high variance in data points, although this is not shown by the error bar? If the variance is high, why were only 6-8 flies tested here? Here, the single data points should be shown and the n number of flies tested for ERG needs to be increased to at least double the number of flies tested.Furthermore, the amplitude of the depolarization of null mutant flies raised in dark for 10 days is lower than the amplitude for the LD flies. Although, in the manuscript, this is described as a "rescued phototransduction". Is the amplitude of "dark flies" consistently lower than the LD flies? More flies (again at least double the number of flies tested before) should be tested here to clarify significance.In general, there is a marked discrepancy for the amplitude between the example data in E and the population data in 1F, both for Vps29 and Vps35 mutants when looking across the three experimental conditions D1-dark, D1 12h-LD, and D10-dark

We thank the reviewers for their careful review of these data, and we apologize for the confusion, which is the result of an unfortunate – and embarrassing – error in compositing of Figure 1F. For our manuscript figures, all graphical elements including labels, axes, and statistical notation (indicating significance) are prepared in Adobe Illustrator, and hyperlinks are embedded to extract the original experimental data charts from the separate application, Graphpad. In our original manuscript submission, these hyperlinks for Figure 1F were incorrect, such that results from the wrong electroretinogram (ERG) experiment were displayed for the data labeled as Day 10 (both 12h-LD and Dark conditions); the Day 1 data is correctly labeled. This error was missed because it was inadvertently introduced in preparing figures for the final submission, but was not present in an earlier circulated version that was reviewed more carefully. This figure “typesetting” error did not impact the results text – which were a valid and correct reporting of all findings.

6) Figure 3A and B: In A, the percentage of mutant/Df flies passing 9 cm in 15 sec is around 40%. While in B, the percentage of null mutant flies (VPS291; nsyb-Gal4) is roughly 10%, although being of the same age as the flies tested in A. How can this significant difference be explained and why weren't the null mutant flies also tested at 2 and 10 days, if they are so different than mutant/Df flies? The authors should test the null mutant flies also for 2 and 10 days to check for any stronger behavior effects compared to the mutant over deficiency flies.

As addressed fully in the response to question #4, above, we have now repeated the locomotor studies including both genotypes (*Vps29^1^*homozygotes and *Vps29^1^ / Df*) and appropriate controls. Our results indeed suggest the presence of a potential modifier, leading to a more severe climbing phenotype in *Vps29^1^* homozygotes; however, our rescue data confirms the specificity of the climbing phenotype for loss of *Vps29*.